# Severity of acute SARS-CoV-2 infection and risk of new-onset autoimmune disease: A RECOVER initiative study in nationwide U.S. cohorts

Shannon Wuller[1]*, Nora G. Singer[2,3], Colby Lewis[4], Elizabeth W. Karlson[5,6,7], Grant S. Schulert[8], Jason D. Goldman[9,10], Jennifer Hadlock[10,11], Jonathan Arnold[12], Kathryn Hirabayashi[13], Lauren E. Stiles[14,15,16], Lawrence C. Kleinman[17,18], Lindsay G. Cowell[19], Mady Hornig[16,20], Margaret A. Hall[21], Mark G. Weiner[4], Michael Koropsak[4], Michelle F. Lamendola-Essel[22], Rachel Kenney[1,23], Richard A. Moffitt[21,24], Sajjad Abedian[25], Shari Esquenazi-Karonika[1], Steven G. Johnson[26], Stephenson Stroebel[4], Zachary S. Wallace[6,27], Karen H. Costenbader[5,6,7] on behalf of the RECOVER Initiative, RECOVER PCORnet EHR Cohort, and the N3C RECOVER EHR Cohort

1 Department of Population Health, New York University Grossman School of Medicine, New York, New York, United States of America, 2 Department of Rheumatology, The MetroHealth System, Cleveland, Ohio, United States of America, 3 Case Western Reserve University School of Medicine, Cleveland, Ohio, United States of America, 4 Department of Population Health Sciences, Weill Cornell Medical College, New York, New York, United States of America, 5 Department of Medicine, Division of Rheumatology, Inflammation and Immunity, Section of Clinical Sciences, Brigham and Women's Hospital, Boston, Massachusetts, United States of America, 6 Rheumatology and Allergy Clinical Epidemiology Research Center, Mongan Institute, Massachusetts General Hospital, Boston, Massachusetts, United States of America, 7 Harvard Medical School, Boston, Massachusetts, United States of America, 8 Department of Pediatrics, University of Cincinnati College of Medicine, Cincinnati Children's Hospital Medical Center, Cincinnati, Ohio, United States of America, 9 Swedish Center for Research and Innovation, Providence Swedish Medical Center, Seattle, Washington, United States of America, 10 University of Washington, Seattle, Washington, United States of America, 11 Institute for Systems Biology, Seattle, Washington, United States of America, 12 Division of General Internal Medicine, University of Pittsburgh School of Medicine, Pittsburgh, Pennsylvania, United States of America, 13 Applied Clinical Research Center, Children's Hospital of Philadelphia, Philadelphia, Pennsylvania, United States of America, 14 Dysautonomia International, East Moriches, New York, United States of America, 15 Stony Brook University Renaissance School of Medicine, Stony Brook, New York, United States of America, 16 RECOVER Patient, Caregiver or Community Representative, United States of America, 17 Department of Pediatrics Child Health Institute of New Jersey, Rutgers Robert Wood Johnson Medical School, New Brunswick, New Jersey, United States of America, 18 Department of Global Public Health, Rutgers School of Public Health, New Brunswick, New Jersey, United States of America, 19 O'Donnell School of Public Health, UT Southwestern Medical Center, Dallas, Texas, United States of America, 20 The Feinstein Institutes for Medical Research, Northwell Health, Manhasset, New York, United States of America, 21 Department of Hematology and Medical Oncology, Emory University, Atlanta, Georgia, United States of America, 22 New York University Grossman School of Medicine, New York, New York, United States of America, 23 Department of Neurology, New York University Grossman School of Medicine, New York, New York, United States of America, 24 Department of Biomedical Informatics, Emory University, Atlanta, Georgia, United States of America, 25 Information Technologies and Services Department, Weill Cornell Medicine, New York, New York, United States of America, 26 Institute for Health Informatics, University of Minnesota, Minneapolis, Minnesota, United States of America, 27 Department of Medicine, Division of Rheumatology, Allergy, and Immunology, Massachusetts General Hospital, Boston, Massachusetts, United States of America

☯ These authors contributed equally to this work.
* Wuller@nyulangone.org

## Abstract

SARS-CoV-2 infection has been associated with increased autoimmune disease risk. Past studies have not aligned regarding the most prevalent autoimmune diseases



**Data availability statement:** PCORnet: Data utilized for this study was obtained from the PCORnet-RECOVER Amazon Warehouse Services (AWS) enclave which is comprised of 40 participating sites from PCORnet (Home - The National Patient-Centered Clinical Research Network ). Please send all data questions or access requests to the corresponding author, who will direct them accordingly. PEDSnet: The results reported in this study are based on detailed individual-level patient data compiled as part of the RECOVER program. Due to the high risk of reidentification based on the number of unique patterns in the date, patient privacy regulations prohibit us from releasing the data publicly. The data are maintained in a secure enclave, with access managed by the program coordinating center to remain compliant with regulatory and program requirements. Please direct requests to access the data, either for reproduction of the work reported here or for other purposes, to the RECOVER EHR Pediatric Coordinating Center (recover@chop.edu). N3C Attribution: The analyses described in this publication were conducted with data or tools accessed through the NCATS N3C Data Enclave https://covid.cd2h.org and N3C Attribution & Publication Policy v 1.2-2020-08-25b supported by NCATS Contract No. 75N95023D00001, Axle Informatics Subcontract: NCATS-P00438-B. This research was possible because of the patients whose information is included within the data and the organizations (https://ncats.nih.gov/research/research-activities/n3c/resources/data-contribution/signatories) and scientists who have contributed to the ongoing development of this community resource [https://doi.org/10.1093/jamia/ocaa196].

**Funding:** This study is part of the NIH Researching COVID to Enhance Recovery (RECOVER) Initiative, which seeks to understand, treat, and prevent the post-acute sequelae of SARS-CoV-2 infection (PASC). For more information on RECOVER, visit https://recovercovid.org/. This research was funded by the National Institutes of Health (NIH) Agreement OTA OT2HL161847 as part of the RECOVER program. Role of the Funder/Sponsor: The funders/sponsors had no role in the design and conduct of the study; collection, management, analysis, and interpretation of the data; preparation, review, or approval of

after infection, however. Furthermore, the relationship between infection severity and new autoimmune disease risk has not been well examined. We used RECOVER's electronic health record (EHR) networks, N3C, PCORnet, and PEDSnet, to estimate types and frequency of autoimmune diseases arising after SARS-CoV-2 infection and assessed how infection severity related to autoimmune disease risk. We identified patients of any age with SARS-CoV-2 infection between April 1, 2020 and April 1, 2021, and assigned them to a World Health Organization COVID-19 severity category for adults or the PEDSnet acute COVID-19 illness severity classification system for children (<age 21). We collected baseline covariates from the EHR in the year pre-index infection date and followed patients for 2 years for new autoimmune disease, defined as ≥ 2 new ICD-9, ICD-10, or SNOMED codes in the same concept set, starting >30 days after SARS-CoV-2 infection index date and occurring ≥1 day apart. We calculated overall and infection severity-stratified incidence ratesper 1000 person-years for all autoimmune diseases. With least severe COVID-19 severity as reference, survival analyses examined incident autoimmune disease risk. The most common new-onset autoimmune diseases in all networks were thyroid disease, psoriasis/psoriatic arthritis, and inflammatory bowel disease. Among adults, inflammatory arthritis was the most common, and Sjögren's disease also had high incidence. Incident type 1 diabetes and hematological autoimmune diseases were specifically found in children. Across networks, after adjustment, patients with highest COVID-19 severity had highest risk for new autoimmune disease vs. those with least severe disease (N3C: adjusted Hazard Ratio, (aHR) 1.47 (95%CI 1.33–1.66); PCORnet aHR 1.14 (95%CI 1.02–1.26); PEDSnet: aHR 3.14 (95%CI 2.42–4.07)]. Overall, severe acute COVID-19 was most strongly associated with autoimmune disease risk in three EHR networks.

## Introduction

The COVID-19 pandemic resulted in significant acute morbidity and mortality from the SARS-CoV-2 virus and has ongoing post-viral consequences. Affecting over 777 million persons as of May 2024, COVID-19 caused over 7 million deaths as assessed by the World Health Organization (WHO) [1]. Among other long-term sequelae of COVID-19, the emergence of clinical autoimmune disease has been reported in many case series and an increasing number of epidemiological studies [2–17].

Collectively, autoimmune diseases affect millions of Americans and are a leading cause of morbidity and mortality [18,19]. Over 100 different autoimmune diseases exist, including type I diabetes, juvenile inflammatory arthritis and inflammatory bowel disease, which predominate among younger people, and rheumatoid arthritis (RA), autoimmune thyroid disease, polymyalgia rheumatica, Sjögren's disease, and varied forms of systemic vasculitis, more common in older people. Most of these related conditions are associated with autoantibody production, targeted or widespread systemic inflammation, and organ damage. They are known to be increasingly prevalent

the manuscript; and decision to submit the manuscript for publication.

**Competing interests:** JH has received funding paid to the Institute of Systems Biology from Bristol Myers Squibb, Gilead, Janssen, Novartis and Pfizer for research unrelated to this study. This does not alter our adherence to PLOS ONE policies on sharing data and materials.

with age and associated with significant morbidity, decreased life expectancy and high medical expenditures [19]. Autoimmune disease pathogenesis is now understood to be due to complex interactions among genetic factors, environmental exposures throughout the lifespan, immune responses to these factors, and interactions between all of these. The environmental triggers of the immune attack on self-tissues characterizing autoimmune diseases are not yet fully elucidated [20]. Autoimmunity arising after viral infections has been widely reported, although it remains challenging to study, with no single viral infection universally accepted as a risk factor for the development of most autoimmune diseases [21].

Recent studies have used administrative databases from the United States (U.S.), Hong Kong, the United Kingdom (U.K.), Germany, and Korea to study the association between SARS-CoV-2 infection and subsequent incident autoimmune diseases in both adults and children [4,15–17,22]. Most of these studies have assessed the risk of developing new autoimmune or autoinflammatory disease among those who tested positive for SARS-CoV-2, compared to the risk among those who tested negative. Using matched cohort designs, risks of a wide range of autoimmune diseases, with variation across studies, have been found to be elevated after SARS-CoV-2 infection compared to those without prior infection [4,15–17,22,23]. A secondary analysis in a study from Hong Kong found that those with more severe hospitalized COVID-19 had increased risk of both transverse myelitis and inflammatory bowel disease, which was not observed when all SARS-CoV-2-infected patients were studied, and that vaccination against SARS-CoV-2 attenuated the overall risk of autoimmune disease after COVID-19 [15]. A recent study using the TriNetX insurance database did not find an influence of vaccination against SARS-CoV-2 on risk of autoimmune disease; however, it reported that those who developed autoimmune disease (vs. those who did not) had an approximately 50% higher risk of having been previously hospitalized with COVID-19 [23].

Given these recent reports and the strong biologic possibility that more severe SARS- CoV-2 infection, with higher circulating concentrations of inflammatory cytokines (e.g., interleukin-6) and cytokine storm syndromes in some, may be more strongly linked to triggering of new autoimmune disease, we aimed to investigate the association between SARS-CoV-2 infection and incident autoimmune diseases [24]. Here, we have investigated COVID-19 severity and the risk of subsequently developing autoimmune disease among adults and children living across the U.S. within the Researching COVID-19 to Enhance Recovery (RECOVER) electronic health record (EHR) networks. RECOVER is an NIH-funded national consortium studying the COVID-19 pandemic's association with long-term health effects. The goal of RECOVER is to improve understanding of and ability to predict, treat, and prevent PASC (post-acute sequelae of SARS-CoV-2, also known as Long COVID). The EHR cohort of RECOVER is comprised of aggregated data from three separate EHR networks, including the National COVID-19 Cohort Collaborative (N3C), the National Patient-Centered Clinical Research Network (PCORnet) and PCORnet's pediatric population, PEDSnet, a pediatric learning system within PCORnet [25–29]. These networks were either established prior to (PCORnet and PEDSnet) or in response to

COVID-19 (N3C). We estimated the type and frequency of incident autoimmune diseases following SARS-CoV-2 infection by severity in the RECOVER EHR cohort, which contains medical data on over 35 million individuals living across the U.S. from April 1, 2019 to March 31, 2023 [23,30].

## Methods

### Data sources

Data from the three RECOVER EHR networks were assembled from over 100 sites across the US. Data included demographics, encounters, medical diagnoses, medications, and selected laboratory results were available. Diagnoses were classified using both the International Classification of Disease (ICD)- 9th and 10th editions codes and the Systematized Nomenclature of Medicine Clinical Terms (SNOMED CT) codes. De-identified EHR data relating to patient encounters from April 1, 2019 to March 31, 2023 were acquired from N3C version 187 from December 2024, PCORnet version S12 from October 2024, and PEDSnet version S9 from July 2023. N3C RECOVER patient selection criteria can be found on the N3C phenotype inclusion criteria website (https://github.com/National-COVID-Cohort-Collaborative/Phenotype_Data_Acquisition/wiki/Latest-Phenotype). The N3C data transfer to NCATS is performed under a Johns Hopkins University Reliance Protocol # IRB00249128 or individual site agreements with NIH. The N3C Data Enclave is managed under the authority of the NIH; information can be found at https://ncats.nih.gov/n3c/resources. The original selection of patients into PCORnet and PEDSnet RECOVER EHR networks are described in S1 Appendix. For PCORnet/PEDSnet, Institute Review Board (IRB) approval was obtained under Biomedical Research Alliance of New York (BRANY) protocol #21-08-508. As part of the Biomedical Research Alliance of New York (BRANY IRB) process, the protocol has been reviewed in accordance with the institutional guidelines. The Biomedical Research Alliance of New York (BRANY) waived the need for consent and HIPAA authorization. Institutional Review Board oversight was provided by the Biomedical Research Alliance of New York, protocol # 21-08-508-380.

### COVID-19 definition and covariates

Patients with evidence of SARS-CoV-2 infection in all three EHR networks were defined and identified for the current study as having a positive SARS-CoV-2 PCR or antigen test, a U07.1 ICD-10 code, a prescription for nirmatrelvir/ritonavir or a medication order for remdesivir between April 1, 2020, and April 1, 2021. A patient's index date was defined as the first indication of SARS-CoV-2 positivity in that time-period. Patients aged 21 and over at their index event were included within N3C and PCORnet analyses, whereas those younger than 21 were included in PEDSnet. Patients were excluded from the study if they had unknown or missing sex. To ensure patients received their healthcare at an in-network site, we required at least one encounter within the 365 days prior to the SARS-CoV-2 infection index date.

The baseline period for collecting study subject characteristics was the year before, up to and including, the SARS-CoV-2 infection index date. Sociodemographic characteristics included: age, sex, race/ethnicity, and medical insurance type. Baseline clinical and behavioral characteristics included: prior history of autoimmune disease, defined as ≥ 1 ICD-9/ICD-10 or SNOMED code for any one autoimmune disease prior to SARS-CoV-2 infection index date (S2–S4 Appendices), healthcare utilization (number of EHR medical encounters) in the 365 days prior to, but not including, the 5 days before index event, the index event time-period (in 3-month intervals), baseline comorbidity scores (by the Charleston Comorbidity Index (CCI) for adults and the Pediatric Medical Complexity Algorithm (PMCAv3.0) for children), ever-smoking status among adults, substance use disorder (S5 Appendix), body mass index (BMI, in $kg/m^2$) categorized according to the World Health Organization (WHO) standards, glucocorticoid use (current vs. non-current oral or intravenous use) (S6 Appendix), and the size of participating RECOVER sites (by pre-inclusion and exclusion criteria patient enrollment numbers in quartiles) [31–34]. Autoimmune rheumatic diseases were excluded from the CCI and the immunological system was excluded from the PMCA.

## COVID-19 severity

For adult patients, we categorized COVID-19 illness severity using encounter location and medical interventions according to the WHO ordinal scale score [35]. Based on encounters within three days before to 16 days after the index event [36], four-level mutually exclusive categories were defined: a) "outpatient with mild conditions" (WHO severity 1–2), if there was no evidence for hospitalization or emergency department encounter; b) "Outpatients with mild conditions with emergency department visit" (WHO severity, approximately 3) if there was an emergency department encounter only; c) "Hospitalized" (WHO severity 4–6) if hospitalized but not critically ill and d) "Hospitalized with ventilation" (WHO severity 7–9) if hospitalized and requiring ICU level care, invasive ventilation, extracorporeal membrane oxygenation (ECMO), or vasopressor/inotropic support. Descriptors here correspond to the labels found in tables and all analyses, i.e., "Outpatient", "Emergency Department", "Hospitalized", and "Hospitalized and on Ventilator". Codes used to identify hospitalized and on ventilator can be found in S7-S8 Appendices.

For pediatric patients, mutually exclusive categories were constructed using PEDSnet's four-level acute COVID-19 severity illness classification. Patients were categorized as asymptomatic, mild, moderate, or severe based on patient COVID-19 symptom clusters identified by using clinically cogent sets of SNOMED CT diagnosis codes examined 7 days before to 13 days after the day of SARS-CoV-2 infection positivity [30]. Asymptomatic patients met no criterion for severity. For more information on the PEDSnet mild – severe severity concepts, see S1 Table.

## Incident autoimmune disease

Patients were followed through their EHRs for up to 24 months after index date with incident autoimmune disease (primary outcome) defined as having ≥ 2 codes within any one new autoimmune disease concept set (condition) starting 30-days or more after index date and occurring ≥ 1 day apart. Autoimmune diseases were identified by ICD-9/ICD-10/SNOMED codes within the same autoimmune disease concept set for 41 distinct autoimmune diseases and one "other" category (S2-S4 Appendices). The minimum 30-day window period after the index date was designed to avoid the capture of prevalent autoimmune diseases and misdiagnoses close to the acute SARS-CoV-2 infection and to compensate for any increased autoimmune disease surveillance bias soon after acute infection. SARS-CoV-2 infection would be unlikely to cause autoimmune disease development within the first month after exposure as most autoimmune diseases develop slowly, with symptoms taking many months to emerge [20]. For individuals with a history of prior autoimmune disease (at least 1 ICD-9/ICD-10 or SNOMED code for any one autoimmune disease prior to SARS-CoV-2 infection index date), incident autoimmune disease in follow-up was included as an outcome only if the diagnosis codes were not within the patient's prior autoimmune disease concept set.

## Statistical analyses

The baseline demographics of the patients in each EHR network were examined with descriptive statistics, overall and according to their COVID-19 illness severity. Unadjusted incidence rates per 1000-person years were calculated overall and by COVID-19 illness severity category for each autoimmune disease concept set. The five most common new autoimmune diseases per 1000 person-years in each EHR network were identified. The association of COVID-19 severity and risk of new autoimmune disease was first assessed using unadjusted Kaplan-Meier curves, censoring for death. We also created unadjusted cumulative incidence curves accounting for the competing risk of death and new autoimmune disease.

We then conducted multivariable-adjusted Cox regression models. In all analyses, the least severe COVID-19 severity category was the reference group. Adjusted hazard ratios (aHR) with 95% confidence intervals (CIs) and corresponding p values were estimated for two primary Cox regression models, censoring for death, end of follow-up, and the outcome of new autoimmune disease. The first model adjusted for age, sex, race, and ethnicity, and the second, fully adjusted model, further adjusted for medical insurance type (as a proxy for socioeconomic status), healthcare

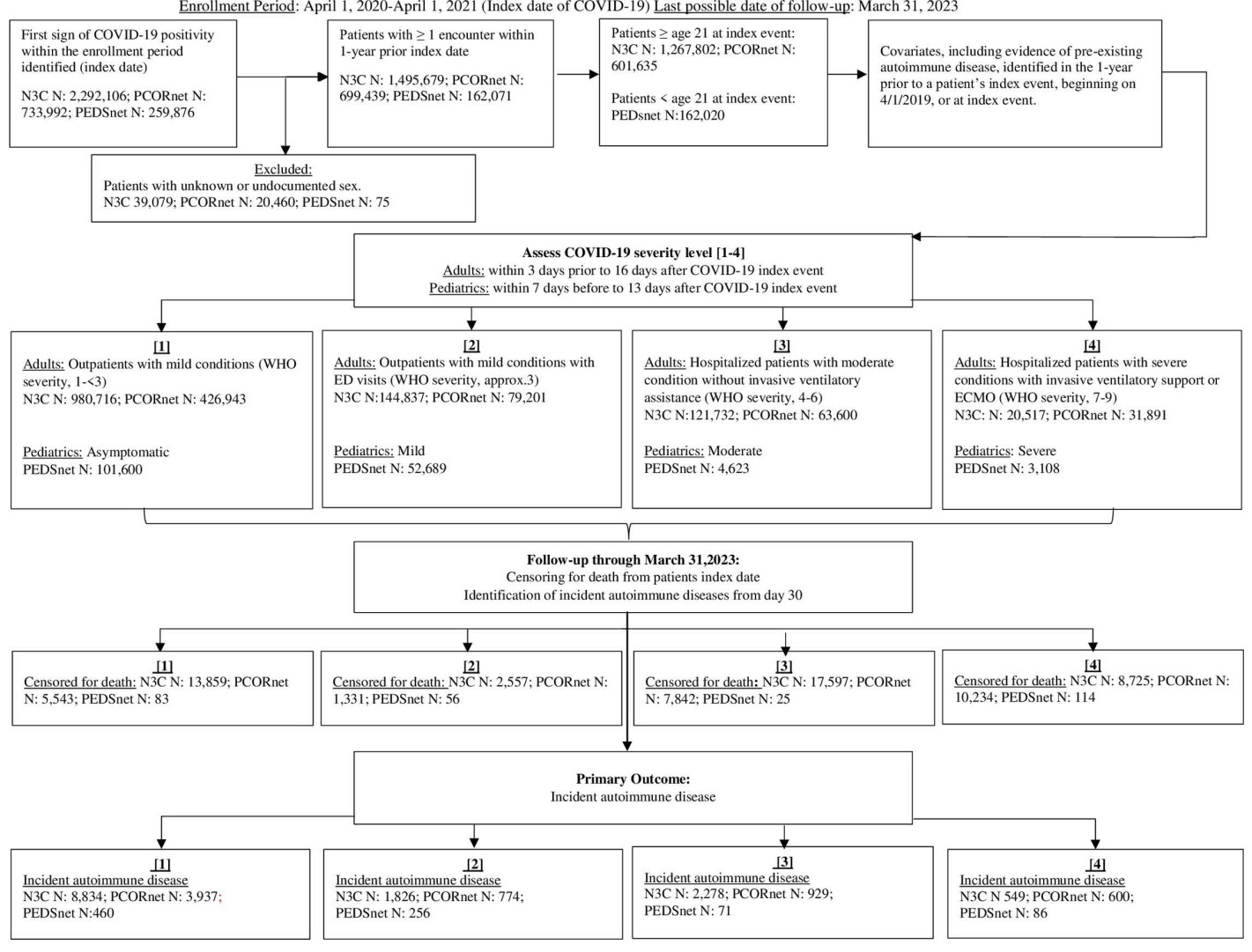

**Fig 1. Flowchart for the selection of COVID-19 patients, COVID-19 severity level assignment, and outcomes.**

utilization, date of SARS-CoV-2 infection, comorbidities, the CCI for adults and PMCA for children, as well as substance use disorder, smoking status (adults only), BMI (according to WHO categories for adults and according to age-based percentiles in children), glucocorticoid use, and RECOVER enrollment site size. We selected our covariates for our adjusted models based on univariable analyses to assess whether they were true confounders (associated with the risk of the predictor and with the risk of the outcome) and also included other variables for face validity (size of the enrollment center, medical insurance, time period of SARS-CoV-2 infection). We tested the proportional hazards assumption by examining the Kaplan-Meier curves and assessing interaction of the predictors with time. Missing data were represented in a missing category. Secondary analyses stratified the fully adjusted models by sex to investigate potential effect modification [37]. In another secondary analysis, we excluded those with a history of any prior autoimmune disease from both the fully adjusted and sex stratified models. Data analyses were performed using R version 4.3.1. R survival and ggsurvfit packages (2023.08.28) [38,39].

**Table 1. Patient characteristics in N3C at time of SARS-CoV-2 infection by illness severity, April 2020 – April 2021.**

| Characteristic | All Patients | Outpatient | Emergency Department | Hospitalized | Hospitalized and on Ventilator |
|---|---|---|---|---|---|
| **N** | 1,267,802 | 980,716 | 144,837 | 121,732 | 20,517 |
| **History of prior autoimmune disease[1]** | 6% | 5% | 6% | 10% | 12% |
| **Mean age (± SD)** | 50 (18) | 48 (17) | 49 (17) | 63 (18) | 61 (17) |
| **Female sex** | 58% | 59% | 61% | 52% | 47% |
| **Race/ethnicity[2]** | | | | | |
| Non-Hispanic Asian | 2% | 2% | 2% | 2% | 3% |
| Non-Hispanic Black | 13% | 10% | 21% | 20% | 21% |
| Non-Hispanic White | 65% | 68% | 55% | 58% | 53% |
| Non-Hispanic Other/Multiple | 3% | 3% | 2% | 3% | 2% |
| Hispanic or Latino | 12% | 12% | 16% | 14% | 16% |
| Missing/Unknown | 5% | 6% | 4% | 4% | 5% |
| **Charlson comorbidity index[3]** | | | | | |
| 0 | 67% | 73% | 60% | 31% | 20% |
| 1-3 | 24% | 21% | 32% | 38% | 35% |
| 4+ | 9% | 5% | 9% | 31% | 44% |
| **Ever smoker** | 6% | 5% | 9% | 10% | 14% |
| **Prior substance abuse** | 6% | 4% | 11% | 13% | 17% |
| **Body mass index[4]** | | | | | |
| Underweight | 1% | 1% | 1% | 2% | 2% |
| Normal weight | 12% | 12% | 12% | 15% | 16% |
| Overweight | 17% | 16% | 19% | 20% | 23% |
| Moderately obese | 13% | 12% | 16% | 16% | 19% |
| Severely obese | 7% | 7% | 10% | 10% | 11% |
| Morbidly obese | 7% | 6% | 10% | 10% | 13% |
| Missing/Unknown | 43% | 47% | 34% | 27% | 15% |
| **Prior glucocorticoid use[5]** | 12% | 10% | 15% | 20% | 24% |
| **Medical insurance** | | | | | |
| Private | 10% | 10% | 10% | 10% | 11% |
| Medicare/Medicaid | 5% | 3% | 9% | 14% | 17% |
| Other | <1% | <1% | 1% | 1% | 1% |
| Missing/Unknown | 84% | 86% | 80% | 75% | 72% |
| **Mean prior encounters (± SD)[6]** | 19 (34) | 17 (30) | 22 (36) | 32 (51) | 36 (57) |
| **Index date period** | | | | | |
| Apr – June 2020 | 10% | 9% | 11% | 17% | 20% |
| Jul – Sep 2020 | 14% | 14% | 14% | 14% | 16% |
| Oct – Dec 2020 | 50% | 51% | 48% | 42% | 37% |
| Jan – Apr 2021 | 26% | 26% | 27% | 27% | 27% |
| **RECOVER site size7** | | | | | |
| 1st quartile | 5% | 4% | 6% | 9% | 10% |
| 2nd quartile | 10% | 9% | 10% | 12% | 18% |
| 3rd quartile | 21% | 21% | 16% | 20% | 25% |
| 4th quartile | 65% | 66% | 68% | 59% | 47% |

Abbreviation: SD, standard deviation.

(1) History of prior autoimmune disease definition: Patients with at least 1 autoimmune disease ICD-9/ICD-10/SNOMED code within any of the autoimmune disease concept set at any point prior to their index date.

*(Continued)*

**Table 1.** (Continued)

(2) Race/ethnicity is commonly inconsistent and/or inaccurately captured in the EHR, especially among minorities, and can lead to significant missing data and misclassification biases [31,32]. Hispanic ethnicity was defined as its own category and not cross-reported with the other race categories for ease of standardization and comparison across a large network of health systems.

(3) Charlson Comorbidity Index is a weighted index classification system using a collection of EHR diagnoses codes from all available data prior to the patient's SARS-CoV-2 infection index to define a patient's risk of mortality [25].

(4) Body Mass Index (BMI in kg/m$^2$) is categorized according to the World Health Organization (WHO) and National Institute of Health (NIH) standards. Categorization BMI cut-offs are as follows: Underweight: < 18.5 kg/m$^2$, normal: 18.5 to 24.9 kg/m$^2$, overweight: 25 to 29.9 kg/m$^2$, moderately obese: 30–34.9 kg/m$^2$, severely obese: 35–39.9 kg/m$^2$, morbidly obese: ≥ 40 [27]. The most recent recorded BMI, either on or up to 1- year prior SARS-CoV-2 infection index event, was used.

(5) Prior glucocorticoid use was defined by having at least 1 prescription for oral or intravenous dexamethasone, betamethasone, prednisolone, methyl-prednisolone, triamcinolone, hydrocortisone, prednisone, or triamcinolone within 1 year to 1 week prior to SARS-CoV-2 infection index event. See S6 Appendix.

(6) Prior encounters were defined as the count of unique days with at least one billing encounter in the 365 days prior to, but not including, the 5 days preceding a patient's SARS-CoV-2 infection. Outpatient and ED visits were assumed to last 1 day.

(7) RECOVER site quartiles are based on overall RECOVER patient enrollment numbers. Sites are categorized by the following; quartile 1: < 140,000 participants, quartile 2: < 220,000 participants, quartile 3: < 400,000 participants, quartile 4: ≥ 400,000 participants.

**Table 2.** Patient characteristics in PCORnet at time of SARS-CoV-2 infection by illness severity, April 2020 – April 2021.

| Characteristic | All Patients | Outpatient | Emergency Department | Hospitalized | Hospitalized and on Ventilator |
|---|---|---|---|---|---|
| **N** | 601,635 | 426,943 | 79,201 | 63,600 | 31,981 |
| **History of prior autoimmune disease[1]** | 8% | 8% | 7% | 11% | 10% |
| **Mean age (± SD)** | 50 (18) | 48 (17) | 48 (17) | 61 (18) | 63 (17) |
| **Female sex** | 55% | 56% | 56% | 50% | 45% |
| **Race/ethnicity[2]** | | | | | |
| Non-Hispanic Asian | 3% | 3% | 2% | 4% | 4% |
| Non-Hispanic Black | 15% | 13% | 23% | 23% | 16% |
| Non-Hispanic White | 46% | 49% | 38% | 42% | 34% |
| Non-Hispanic Other/Multiple | 5% | 6% | 5% | 6% | 4% |
| Hispanic or Latino | 15% | 13% | 20% | 21% | 13% |
| Missing/Unknown | 16% | 18% | 12% | 5% | 29% |
| **Charlson comorbidity index[3]** | | | | | |
| 0 | 60% | 68% | 61% | 2745% | 26% |
| 1-3 | 27% | 24% | 30% | 39% | 38% |
| 4+ | 12% | 8% | 9% | 34% | 36% |
| **Ever smoker** | 22% | 20% | 27% | 35% | 20% |
| **Prior substance abuse** | 6% | 4% | 10% | 14% | 11% |
| **Body mass index[4]** | | | | | |
| Underweight | 1% | 1% | 1% | 1% | 1% |
| Normal weight | 11% | 11% | 8% | 10% | 9% |
| Overweight | 15% | 16% | 13% | 16% | 14% |
| Moderately obese | 12% | 12% | 11% | 13% | 11% |
| Severely obese | 7% | 6% | 7% | 8% | 6% |
| Morbidly obese | 6% | 5% | 7% | 8% | 6% |
| Missing/Unknown | 49% | 49% | 53% | 45% | 52% |
| **Prior glucocorticoid use[5]** | 10% | 9% | 8% | 12% | 12% |
| **Medical insurance** | | | | | |

*(Continued)*

**Table 2.** (Continued)

| Characteristic | All Patients | Outpatient | Emergency Department | Hospitalized | Hospitalized and on Ventilator |
|---|---|---|---|---|---|
| *Private* | 25% | 29% | 21% | 15% | 8% |
| *Medicare/Medicaid* | 13% | 11% | 17% | 23% | 20% |
| *Other* | 2% | 2% | 1% | 2% | 1% |
| *Missing/Unknown* | 60% | 59% | 61% | 60% | 71% |
| ***Mean prior encounters (± SD)[6]*** | 12 (19) | 11 (16) | 11 (17) | 19(28) | 21 (29) |
| ***Index date period*** | | | | | |
| *Apr – June 2020* | 19% | 19% | 16% | 22% | 32% |
| *Jul – Sep 2020* | 19% | 21% | 16% | 15% | 12% |
| *Oct – Dec 2020* | 35% | 36% | 35% | 34% | 25% |
| *Jan – Apr 2021* | 26% | 25% | 33% | 29% | 30% |
| ***RECOVER site size7*** | | | | | |
| *1st quartile* | 1% | 1% | 2% | 1% | 1% |
| *2nd quartile* | 9% | 7% | 15% | 14% | 3% |
| *3rd quartile* | 15% | 17% | 11% | 9% | 19% |
| *4th quartile* | 75% | 76% | 72% | 76% | 77% |

Abbreviation: SD, standard deviation.

(1) History of prior autoimmune disease definition: Patients with at least 1 autoimmune disease ICD-9/ICD-10/SNOMED code within any of the autoimmune disease concept set at any point prior to their index date.

(2) Race/ethnicity is commonly inconsistent and/or inaccurately captured in the EHR, especially among minorities, and can lead to significant missing data and misclassification biases [31,32]. Hispanic ethnicity was defined as its own category and not cross-reported with the other race categories for ease of standardization and comparison across a large network of health systems.

(3) Charlson Comorbidity Index, CCI, is a weighted index classification system using a collection of EHR diagnoses codes from all available data prior to the patient's SARS-CoV-2 infection index to define a patient's risk of mortality [25].

(4) Body Mass Index (BMI, in kg/m$^2$) is categorized according to the World Health Organization (WHO) and National Institute of Health (NIH) standards. Categorization BMI cut offs are as follows: Underweight: < 18.5 kg/m$^2$, normal: 18.5 to 24.9 kg/m$^2$, overweight: 25 to 29.9 kg/m$^2$, moderately obese: 30–34.9 kg/m$^2$, severely obese: 35–39.9 kg/m$^2$, morbidly obese: ≥ 40 [27]. The most recent recorded BMI, either on or up to 1- year prior SARS-CoV-2 infection index event, was used.

(5) Prior glucocorticoid use was defined by having at least 1 prescription for oral or intravenous dexamethasone, betamethasone, prednisolone, methylprednisolone, triamcinolone, hydrocortisone, prednisone, or triamcinolone within 1 year to 1 week prior to SARS-CoV-2 infection index event. See S6 Appendix.

(6) Prior encounters were defined as the count of unique days with at least one billing encounter in the 365 days prior to, but not including, the 5 days preceding a patient's SARS-CoV-2 infection. Outpatient and ED visits were assumed to last 1 day.

(7) RECOVER site quartiles are based on overall RECOVER patient enrollment numbers. Sites are categorized by the following; quartile 1: < 140,000 participants, quartile 2: < 220,000 participants, quartile 3: < 400,000 participants, quartile 4: ≥ 400,000 participants.

## Results

### Study populations

Fig 1 shows the flow diagram of the cohort construction and overall outcomes of the analysis.

Patient characteristics by COVID-19 severity status for each EHR network are shown in Tables 1–3. The following number of patients with SARS-CoV-2 infection were identified between April 1, 2020, and April 1, 2021: 1,267,802 adults within N3C, 601,635 adults within PCORnet, and 162,020 pediatric patients in PEDSnet. For both N3C and PCORnet, mean patient age was 50 (standard deviation, SD, 18) and increased as COVID-19 severity increased, with those in the highest severity level being 61-63 (SD 17) years old. For PEDSnet, the overall mean age was 12 (SD 7) and more consistent across the severity categories.

**Table 3. Patient characteristics in PEDSnet at time of SARS-CoV-2 infection by illness severity, April 2020 – April 2021.**

| Characteristic | All Patients | Asymptomatic | Mild | Moderate | Severe |
|---|---|---|---|---|---|
| **N** | 162,020 | 101,600 | 52,689 | 4,623 | 3,108 |
| **History of prior autoimmune disease[1]** | 3% | 3% | 3% | 9% | 11% |
| **Mean age (± SD)** | 12 (7) | 12 (6) | 12 (7) | 11 (7) | 11 (7) |
| **Female sex** | 51% | 51% | 52% | 51% | 47% |
| **Race/ethnicity[2]** | | | | | |
| Non-Hispanic Asian | 3% | 3% | 3% | 2% | 3% |
| Non-Hispanic Black | 14% | 14% | 14% | 17% | 24% |
| Non-Hispanic White | 38% | 38% | 38% | 33% | 31% |
| Non-Hispanic Other/Multiple | 2% | 2% | 1% | 2% | 2% |
| Hispanic or Latino | 29% | 28% | 31% | 36% | 31% |
| Missing/Unknown | 14% | 15% | 13% | 9% | 9% |
| **Pediatric medical complexity algorithm: complex/chronic[8]** | | | | | |
| Neither chronic nor complex | 90% | 91% | 90% | 76% | 57% |
| Chronic | 4% | 4% | 5% | 6% | 4% |
| Complex-chronic | 6% | 5% | 5% | 18% | 39% |
| **Prior substance abuse** | 2% | 2% | 2% | 4% | 6% |
| **Body mass index percentile[4]** | | | | | |
| <5th percentile | 2% | 2% | 3% | 5% | 6% |
| 5th – 84th percentile | 22% | 22% | 21% | 29% | 31% |
| 85th – 94th percentile | 6% | 6% | 6% | 9% | 9% |
| > 95th percentile | 19% | 16% | 23% | 18% | 22% |
| Missing/Unknown | 51% | 53% | 48% | 39% | 31% |
| **Prior glucocorticoid use[5]** | 8% | 6% | 8% | 33% | 46% |
| **Medical insurance** | | | | | |
| Private | 14% | 17% | 10% | 13% | 19% |
| Medicare/Medicaid | 10% | 10% | 7% | 20% | 25% |
| Other | <1% | <1% | <1% | <1% | <1% |
| Missing/Unknown | 76% | 73% | 83% | 67% | 55% |
| **Mean prior encounters[6] (± SD)** | 7 (14) | 6 (12) | 7 (12) | 15 (28) | 23 (39) |
| **Index date period** | | | | | |
| Apr – June 2020 | 8% | 6% | 9% | 12% | 17% |
| Jul – Sep 2020 | 19% | 18% | 21% | 21% | 19% |
| Oct – Dec 2020 | 41% | 42% | 39% | 32% | 32% |
| Jan – Mar 2021 | 32% | 33% | 32% | 34% | 32% |
| **RECOVER site size[7]** | | | | | |
| 1st quartile | 10% | 10% | 9% | 9% | 9% |
| 2nd quartile | 18% | 17% | 20% | 19% | 20% |
| 3rd quartile | 18% | 19% | 17% | 24% | 29% |
| s4th quartile | 54% | 54% | 55% | 48% | 42% |

Abbreviation: SD, standard deviation.

(1) History of prior autoimmune disease definition: Patients with at least 1 autoimmune disease ICD-9/ICD-10/SNOMED code within any of the autoimmune disease concept set at any point prior to their index date.

(2) Race/ethnicity is commonly inconsistent and/or inaccurately captured in the EHR, especially among minorities, and can lead to significant missing data and misclassification biases [31,32]. Hispanic ethnicity was defined as its own category and not cross-reported with the other race categories for ease of standardization and comparison across a large network of health systems.

*(Continued)*

**Table 3.** (Continued)

(4) Body Mass Index (BMI in kg/m²) is categorized according to the World Health Organization (WHO) and National Institute of Health (NIH) standards. The most recent recorded BMI, either on or up to 1- year prior to the SARS-CoV-2 infection index event, was used.

(5) Prior glucocorticoid use was defined by having at least 1 prescription for oral or intravenous dexamethasone, betamethasone, prednisolone, methylprednisolone, triamcinolone, hydrocortisone, prednisone, or triamcinolone within 1 year to 1 week prior to SARS-CoV-2 infection index event. See S6 Appendix.

(6) Prior encounters were defined as the count of unique days with at least one billing encounter in the 365 days prior to, but not including, the 5 days preceding a patient's SARS-CoV-2 infection. Outpatient and ED visits were assumed to last 1 day.

(7) RECOVER site quartiles are based on overall RECOVER patient enrollment numbers. Sites are categorized by the following; quartile 1: < 140,000 participants, quartile 2: < 220,000 participants, quartile 3: < 400,000 participants, quartile 4: ≥ 400,000 participants.

(8) Pediatric medical complexity algorithm (PMCA version 3.0): complex/chronic is an algorithm to define pediatric chronic disease and medical complexity using patients' previous EHR diagnosis codes within the 3 years before the SARS-CoV-2 infection index event. The most conservative definition of the algorithm was used [33].

A higher proportion of patients within the most severe COVID-19 level were male (male: N3C 53%; PCORnet 55%; PEDSnet 53%) and those with high disease severity also had more prevalent pre-existing autoimmune disease (N3C 12%, PCORnet 10%, PEDSnet 11%). Most patients were identified as Non-Hispanic White (N3C 65%; PCORnet 46%; PEDSnet 38%). However, compared to the least COVID-19 severity levels, the more severe COVID-19 severity levels contained higher percentages of people who were identified as Non-Hispanic Blacks and/or Hispanic Latinos. Although most adults, 65–67%, and 90% of children had no indications of past comorbidities, as indicated by CCI or PMCA, a higher proportion of patients with CCI of ≥4 or PMCA complex/chronic were found among those who had more severe COVID-19. Even though trends across CCI scores and COVID-19 severity levels were similar between the adult networks, N3C had a greater number of patients with CCI of ≥4 in hospitalized and on ventilator than PCORnet (44% vs 36%, respectively). A small percentage of patients had a prior history of substance abuse or glucocorticoid use. Across all networks, the percentages of both variables increased as COVID-19 severity levels increased, and this trend was most evident in PEDSnet where 46% of those with severe COVID-19 illness had a prior history of glucocorticoid usage.

Among the patients in the N3C network, there was evidence of more healthcare utilization in the year prior to their SARS-CoV-2 infection than there was among the PCORnet or PEDSnet patients (19 (SD 34) vs. 12 (SD 19) and 7 (SD 14) healthcare encounters respectively). Across networks, higher COVID-19 severity was associated with increased incidence of prior healthcare utilization.

Across the three networks, more SARs-CoV-2 infections occurred between October- December 2020 than during other time periods. All study populations were primarily drawn from larger RECOVER sites (N3C 65%; PCORnet 75%; PEDSnet 54%), although the range of COVID-19 severity levels was seen at all RECOVER sites.

## Risk of autoimmune disease

Table 4 and Figs 2–4 display unadjusted incidence rates per 1000 person-years and Kaplan-Meier curves for autoimmune disease incidence by COVID-19 severity levels for each EHR network, starting > 30 days after the index date. Within N3C and PEDSnet, unadjusted overall autoimmune disease incidence per 1000 person-years increased as COVID-19 severity levels increased. However, within PCORnet, unadjusted autoimmune disease incidence was similar across the COVID-19 severity categories. Cumulative incidence analyses accounting for the competing risk of death obtained similar results as the Kaplan-Meier models (S1-S3 Figs).

The most common incident autoimmune diseases arising in the those with evidence of SARS-CoV-2 infection, as indicated by incidence per 1000 person-years, aligned in the two adult networks; the pediatric population had notable similarities to the adults. Thyroid disease, psoriasis or psoriatic arthritis, and inflammatory bowel disease were consistent across all 3 EHR networks. Among adults, inflammatory arthritis was the most common autoimmune disease arising from

**Table 4. Incidence rates per 1000 person-years for any new autoimmune disease and each EHR network's five most prevalent autoimmune diseases for all COVID-19 patients, stratified by WHO severity categories[9], starting > 30 days after SARS-CoV-2 infection index date.**

| | Most Common New Autoimmune Diseases | All Patients (n = 1,267,802) | Outpatient (n = 980,716) | Emergency Department (n = 144,837) | Hospitalized (n = 121,732) | Hospitalized and on Ventilator (n = 20,517) |
|---|---|---|---|---|---|---|
| **N3C** | Any new autoimmune disease | 7.77 | 7.09 | 8.69 | 12.20 | 14.37 |
| | #1 Inflammatory arthritis or RA | 1.41 | 1.22 | 1.67 | 2.70 | 2.77 |
| | #2 Psoriasis or psoriatic arthritis | 1.32 | 1.29 | 1.38 | 1.60 | 1.67 |
| | #3 Hashimoto thyroiditis | 1.07 | 1.10 | 1.00 | 0.90 | 0.85 |
| | #4 Inflammatory bowel disease | 0.75 | 0.69 | 0.89 | 1.10 | 1.07 |
| | #5 Sjögren's disease | 0.61 | 0.57 | 0.76 | 0.80 | 1.01 |
| **PCORnet** | **Most Common New Autoimmune Diseases** | **All Patients (n = 601,635)** | **Outpatient (n = 426,943)** | **Emergency Department (n = 79,201)** | **Hospitalized (n = 63,600)** | **Hospitalized and on Ventilator (n = 31,981)** |
| | Any new autoimmune disease | 4.27 | 4.32 | 4.00 | 4.39 | 4.14 |
| | #1 Inflammatory arthritis or RA | 0.79 | 0.78 | 0.87 | 0.72 | 0.80 |
| | #2 Sjögren's disease | 0.72 | 0.74 | 0.74 | 0.63 | 0.53 |
| | #3 Hashimoto thyroiditis | 0.62 | 0.69 | 0.61 | 0.32 | 0.38 |
| | #4 Psoriasis or psoriatic arthritis | 0.61 | 0.65 | 0.50 | 0.51 | 0.44 |
| | #5 Inflammatory bowel disease | 0.48 | 0.52 | 0.41 | 0.32 | 0.50 |
| **PEDSnet** | **Most Common New Autoimmune Diseases** | **All Patients (n = 162,020)** | **Asymptomatic (n = 101,600)** | **Mild (n = 52,689)** | **Moderate (n = 4,623)** | **Severe (n = 3,108)** |
| | Any new autoimmune disease | 2.71 | 2.27 | 2.44 | 7.80 | 14.50 |
| | #1 Thyroid disease | 0.33 | 0.27 | 0.38 | 0.65 | 0.99 |
| | #2 Inflammatory bowel disease | 0.32 | 0.32 | 0.22 | 1.20 | 0.50 |
| | #2 Type 1 diabetes | 0.32 | 0.27 | 0.26 | 0.98 | 2.49 |
| | #4 Hematological: other or unspecified | 0.28 | 0.24 | 0.22 | 0.98 | 1.82 |
| | #5 Psoriasis or psoriatic arthritis | 0.23 | 0.18 | 0.30 | 0.43 | 0.33 |

Abbreviation: WHO, World Health Organization.

(9) World Health Organization COVID-19 categorization [35].

SARS-CoV-2 infection, and Sjögren's disease also had high incidence. Incident type 1 diabetes and hematological autoimmune diseases were specifically found in the pediatric population. A complete list of incidence rates per 1000 –person-years among the overall population for any new autoimmune disease by EHR network can be found in S9 Appendix.

Unlike in N3C and PEDsnet, the unadjusted incidence of autoimmune disease in PCORnet was lowest in the ED level of severity, followed by those hospitalized with ventilation category; it was highest in the hospitalized without ventilation and outpatient categories. In PEDSnet, the largest difference in incidence rates between severity levels was observed for the risks of type 1 diabetes among those with moderate vs. severe COVID-19 (IR 0.98 vs. 2.49 per 1000 person-years).

In our univariable analyses (data not shown), we found age, sex, race, ethnicity, obesity, smoking, and comorbidities to be true confounders (associated with risk of infection and risk of autoimmune disease) and thus adjusted for them in our models. The results of the adjusted Cox regression models for risk of developing a new autoimmune disease > 30 days after SARS-CoV-2 infection index in each of the three networks, according to COVID-19 severity, are shown in Table 5, S4 Fig, and S2-S4 Tables. In N3C, PCORnet, and PEDSnet, patients in the highest COVID-19 severity levels had increased risk of any new autoimmune disease compared to those in the least severe COVID-19 levels after adjusting for age, sex, and race [aHR 2.09 (95% CI 1.85–2.36); aHR 1.29 (95% CI 1.17–1.40); aHR 6.93 (95% CI 5.49–8.73), respectively]. These findings persisted in the fully adjusted models [N3C: aHR 1.47 (95% CI 1.33–1.66); PCORnet: aHR 1.14 (95% CI 1.02–1.26); PEDSnet: aHR 3.14 (95% CI 2.42–4.07)].

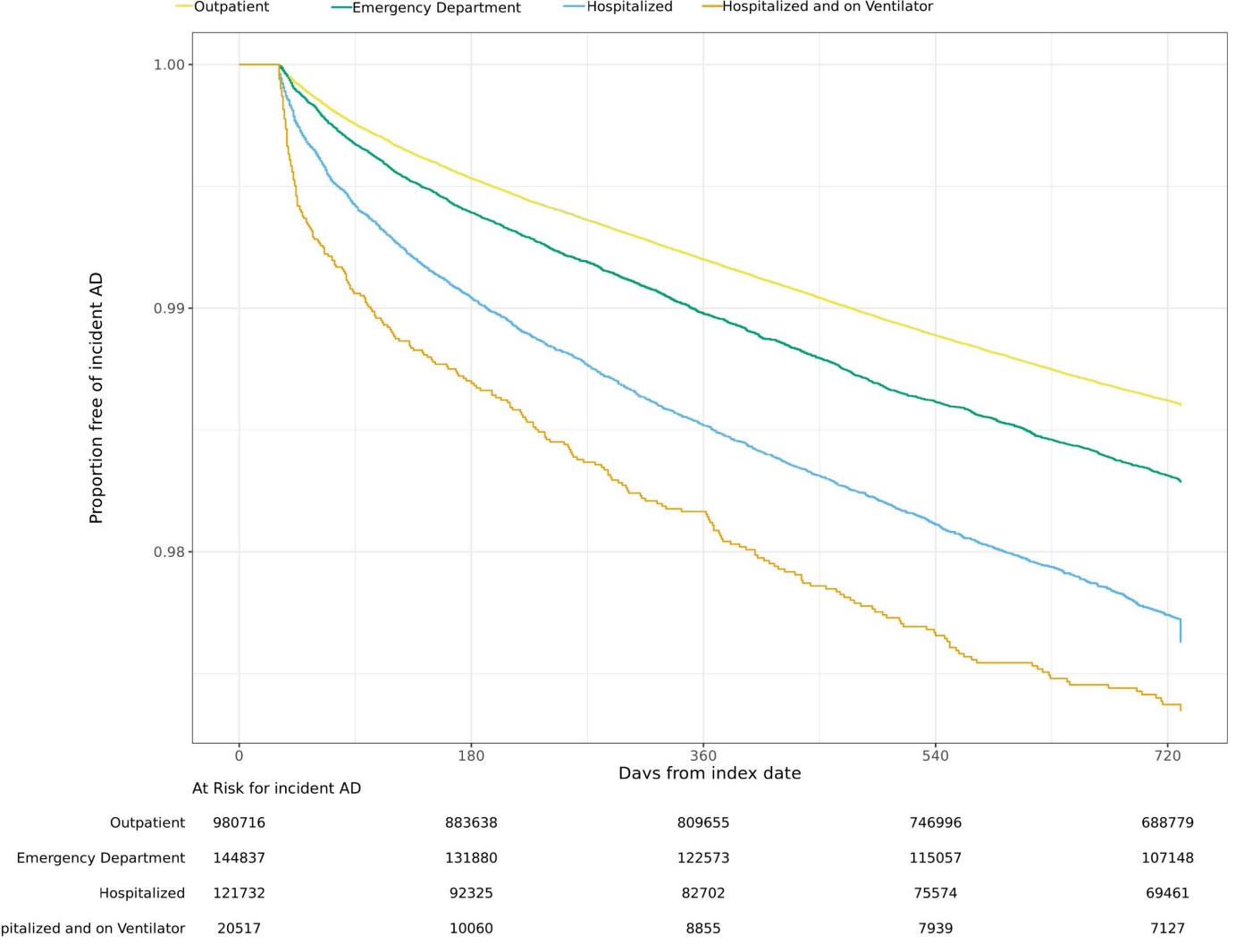

**Fig 2. N3C Kaplan-Meier curves for any new incident autoimmune disease.** N3C Kaplan-Meier survival analysis comparing any autoimmune disease-free survival probabilities by COVID-19 severity level by time in days.

In N3C, those hospitalized with ventilation had elevated risks for new autoimmune disease compared to outpatients, as did those within the ED or hospitalized group [aHR 1.13 (95% CI 1.07–1.18), aHR 1.33 (95% CI 1.26–1.41), respectively]. In contrast, PCORnet patients hospitalized without ventilation had a 18% reduction in risk when compared to outpatients [aHR 0.82 (95%CI 0.73–0.90)]. In PEDsnet, risk of new autoimmune disease increased with increasing COVID-19 severity levels in all models, but in only the moderate and severe categories were risks significantly elevated compared to those in the asymptomatic infection category. In addition to those with severe infections, children with moderate COVID-19 had a 2-fold increase in autoimmune disease risk after full adjustment [aHR 2.15 (95% CI 1.65–2.79)].

## Secondary analyses

When excluding patients with a prior history of any autoimmune disease, associations of COVID-19 severity with the risk of autoimmune disease persisted, but slightly attenuated, among those in the emergency department or hospitalized severity levels within N3C and moderate or severe levels in PEDSnet [N3C aHR 1.11 (95% CI 1.05–1.17), aHR 1.19 (95%

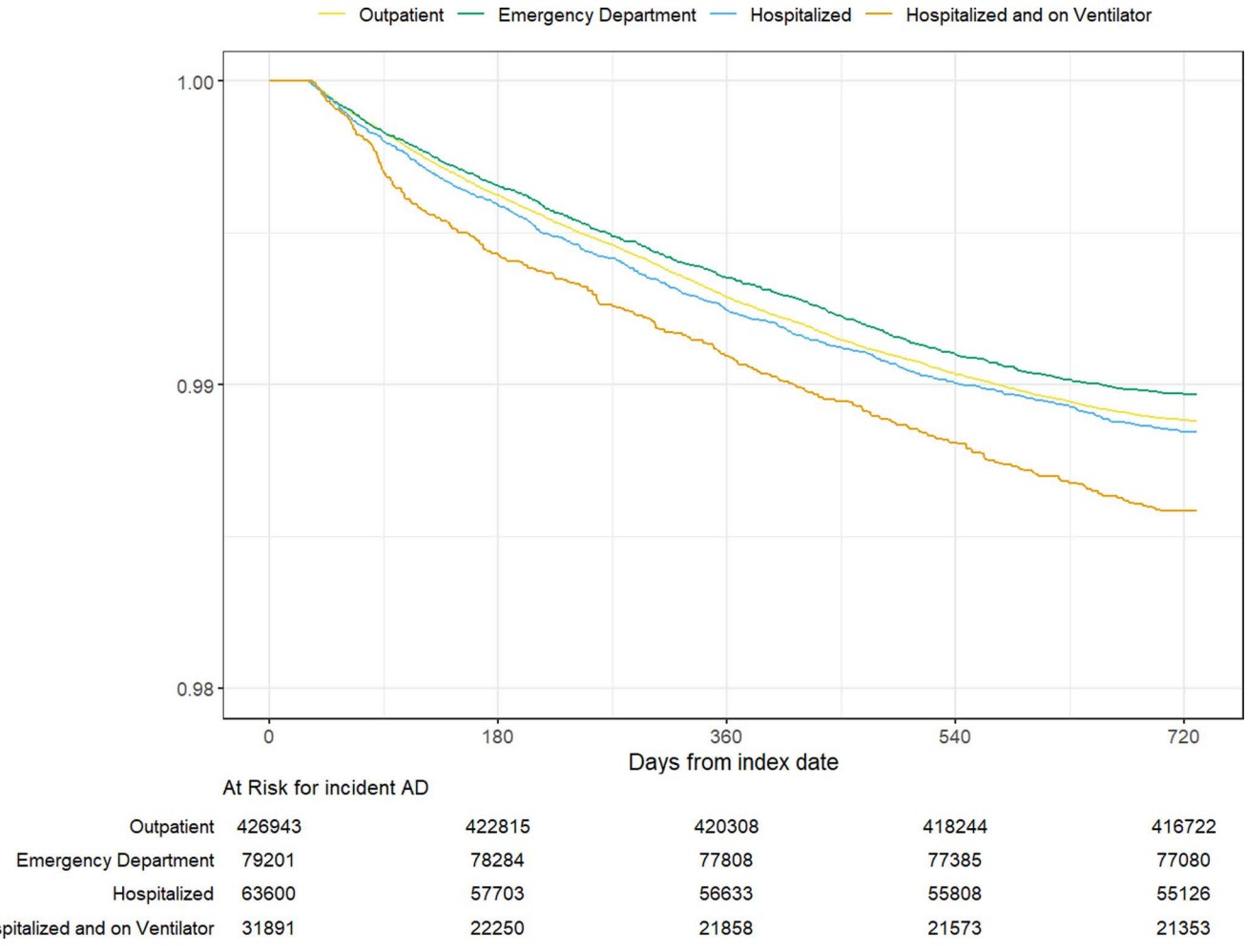

**Fig 3. PCORnet Kaplan-Meier curves for any new incident autoimmune disease.** PCORnet Kaplan-Meier survival analysis comparing any autoimmune disease-free survival probabilities by COVID-19 severity level by time in days.

CI 1.12–1.27),respectively; PEDSnet aHR 2.19 (95% CI 1.64–2.93), aHR 3.29 (95% CI 2.47–4.39), respectively]. In N3C and PCORnet, those hospitalized with ventilation had an increased, but non-significant, risk of autoimmune disease [aHR 1.15 (95% CI 0.98–1.35), aHR 1.13 (95% CI 0.99–1.27), respectively]. Sex stratification of all fully-adjusted models, models with and excluding those with prior autoimmune disease, did not reveal any significant difference by sex in all three EHR networks.

## Discussion

In this large US-based EHR study, we examined the association of COVID-19 severity with the risk of new autoimmune disease diagnosis. Our findings suggest that the risk of being diagnosed with an autoimmune disease after SARS-CoV-2 infection was highest among those who experienced the most severe COVID-19, i.e., those who were hospitalized and ventilated. These findings were observed in three different U.S.-based networks, including one in a pediatric population. Our analysis corroborates and extends findings from recent studies which found that more severe COVID-19 illness might be associated with increased subsequent autoimmune disease risk [15,23,40].

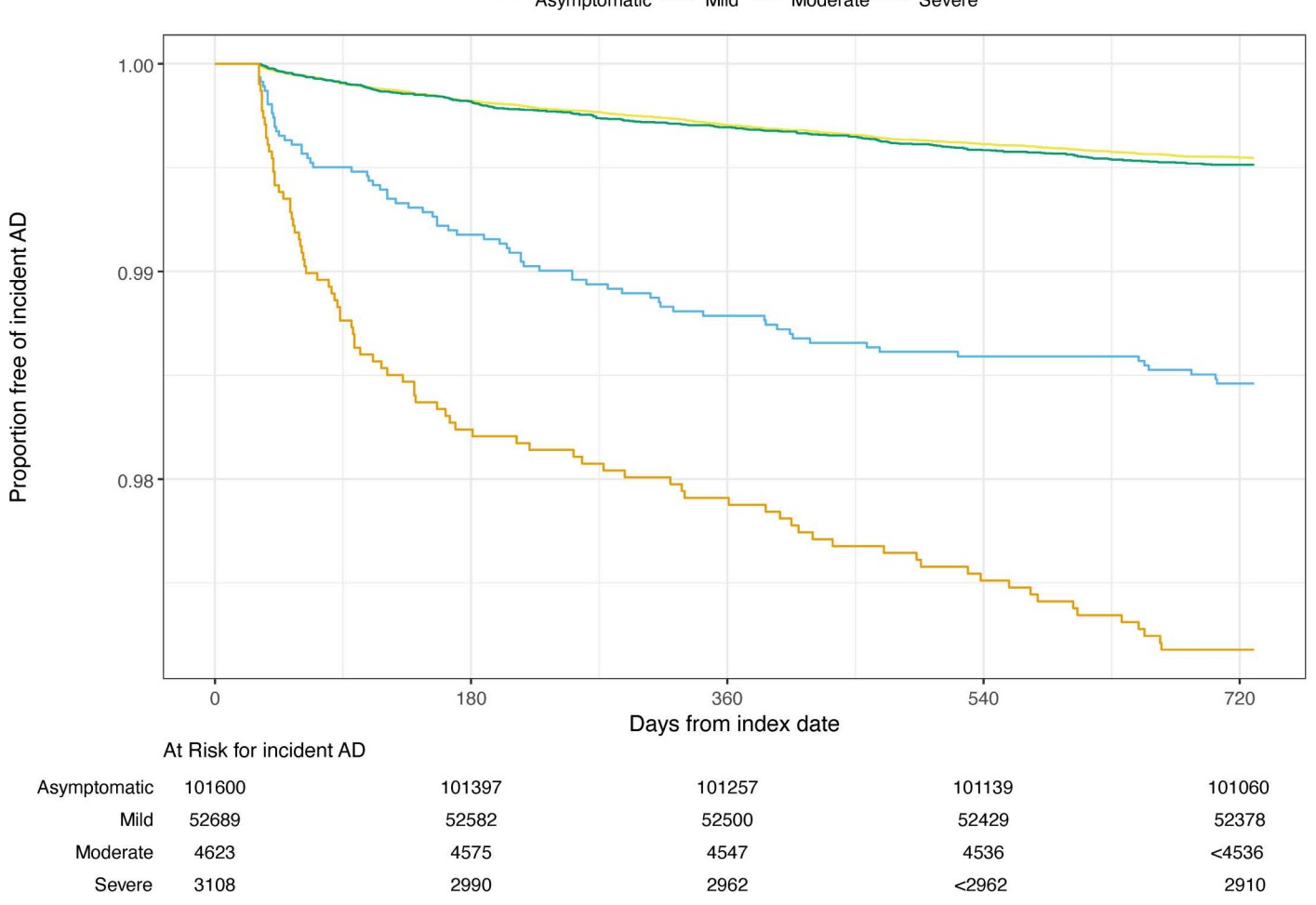

**Fig 4. PEDSnet Kaplan-Meier curves for any new incident autoimmune disease.** PEDSnet Kaplan-Meier survival analysis comparing any autoimmune disease-free survival probabilities by COVID-19 severity level by time in days. Small cell sizes (<20) are suppressed for patient privacy.

In these three networks, the most common adult autoimmune diseases newly diagnosed after SARS-CoV-2 infection shared by both adult and pediatric networks were autoimmune thyroiditis (Hashimoto's in adults), psoriasis or psoriatic arthritis, and inflammatory bowel disease. In the pediatric population, the third most common autoimmune disease was type I diabetes, which has also been reported after other viral infections such as mumps, parainfluenza, human herpes virus 6, enteroviruses, and Coxsackievirus B1 (CVB1) serotype among children [41,42]. In the adult populations, inflammatory arthritis/rheumatoid arthritis (RA) was most common. This was also reported in a study in a Colombian cohort in which the incidence rate ratio for rheumatoid arthritis following SARS-CoV-2 infection was 2 [43]. The alignment of these results across different networks and studies is important as it reinforces the consistency and reliability of the findings.

Past studies have reported increases in the risk of developing systemic vasculitis after SARS-CoV-2 infection as among the most common incident autoimmune disease sequalae [4,17,22,23]. Four of five large recently published studies on this topic employed only one billing code to define incident autoimmune disease, which can be highly non-specific, particularly in the setting of the types of vague and ongoing constitutional symptoms that emerged after SARS-CoV-2 infection. Our study required a more specific definition, two separate encounters with similar autoimmune ICD9, ICD10 or SNOMED

**Table 5. Multivariable adjusted hazard ratios for incident autoimmune disease by COVID-19 severity categories for N3C, PCORnet, and PEDS-net from day > 30 to end of study period.**

| | Outcomes | Outpatient aHR (95% CI) | Emergency Department aHR (95% CI) | Hospitalized aHR (95% CI) | Hospital-ized and on ventilator aHR (95% CI) |
|---|---|---|---|---|---|
| **N3C** | **Model 1:** age, sex, race | REF | 1.24 (1.19-1.30) | 1.69(1.61-1.78) | 2.09(1.85-2.36) |
| | **Model 2:** full model[10] | REF | 1.13(1.07-1.18) | 1.33(1.26-1.41) | 1.47(1.33-1.66) |
| **PCORnet** | **Model 1:** age, sex, race | REF | 0.91(0.84-0.99;) | 0.91(0.83-0.99) | 1.29(1.17-1.40) |
| | **Model 2:** full model[10] | REF | 0.95(0.88-1.03) | 0.82(0.73-0.90) | 1.14(1.02-1.26) |
| **PEDSnet** | | Asymptom-atic aHR (95% CI) | Mild aHR (95% CI) | Moderate aHR (95% CI) | Severe aHR (95% CI) |
| | **Model 1:** age, sex, race | REF | 1.07 (0.92-1.25) | 3.65 (2.84-4.69) | 6.93 (5.49-8.73) |
| | **Model 2:** full model[10] | REF | 1.04 (0.89-1.22) | 2.15(1.65-2.79) | 3.14 (2.42-4.07) |

Abbreviation: aHR (95%CI), adjusted Hazard ratio with 95% confidence interval.

(10) Full model adjusted for age, sex, race, medical insurance type, healthcare utilization, date of SARS-CoV-2 infection, comorbidities (CCI for adults, PMCA for children), substance use disorder, smoking status (adults only), body mass index, glucocorticoid use, and RECOVER enrollment site size.

codes recorded at least 1 day apart to improve specificity and positive predictive value [44]. Some autoimmune diseases, such as vasculitis, may have been overlooked as they may have been self-limited (e.g., Henoch-Schoenlein purpura in children, or leukocytoclastic vasculitis). In past studies, systemic lupus erythematosus (SLE) and Sjögren's disease were reported to be common post-SARS-CoV-2 infection [4,16]. In the current study, we found that autoimmune hemolytic anemia and Evan's syndrome, which can precede childhood SLE, were among the five most common PEDSnet autoimmune diseases associated with SARS-CoV-2 infection.

The association of viral infection severity with increased risk of autoimmune disease is important and biologically plausible. It may be related to cytokine release syndrome seen in severe cases [45,46]. While influenza A and other viral infections induce release of many inflammatory cytokines, coronaviruses in general and SARS-CoV-2 in particular, have been shown to have very strong stimulatory influences on immune and inflammatory cytokines, such as type I interferons (IFNs), key anti-viral cytokines limiting viral infection via activation of the innate immune system [46,47]. Neutrophil extracellular traps (NETs), which stimulate production of type I IFNs, have been identified in severe COVID-19 illness and additionally are implicated in autoimmune disease pathogenesis. NETs prime dendritic cells to present their nuclear contents, stimulating B cell production and affinity maturation along with formation and release of autoantibodies [48,49]. Thus, host immune responses developed to protect from viral infection, when not appropriately downregulated, may contribute to increased risk of developing autoimmune disease, including in the context of viral infections.

## Strengths and limitations

Our study has several strengths including large nationwide EHR networks, the inclusion of a pediatric population, the requirement for two independent billing codes on separate days for a new autoimmune diagnosis (increasing AD diagnosis positive predictive value) [44], and secondary analyses excluding those with pre-existing autoimmune disease of any kind. The simultaneous analysis of the three EHR networks, which contain only 20 overlapping sites with each other, not only broadened the scope of our analysis, but also enabled us to establish a more definitive connection between the severity of COVID-19 requiring ICU admission/ventilation, compared to less severe COVID-19, and the subsequent increase in incident autoimmune disease over a two-year period.

Of recent studies of autoimmune disease arising after SARS-CoV-2 infection, only one has included children [17]. Our inclusion of a large pediatric population provides a more comprehensive understanding of the disease's impact on this

critical demographic, with its susceptibility to a different spectrum of autoimmune diseases. Given the genetic predisposition that increases the likelihood of those with existing autoimmune disease to develop a new AD, we included all participants in our primary analyses and applied a stricter incident AD definition to those with a history of prior AD. In our secondary analyses excluding all patients with evidence of any autoimmune disease in the baseline period, we obtained similar results as our primary analysis. Although risk for AD fell just short of significance in N3C and PCORnet hospitalized with ventilation patients, we still believe that this association is plausible. The change in significance is likely due to inherent EHR limitations (e.g., patients interacting with the same healthcare system in follow up or potential misclassified in COVID-19 severity categories), as PEDSnet and other N3C severity levels showed increased risk for AD that remained significant. However, we acknowledge that this still requires further investigation as the established link between severe COVID-19 and autoimmune disease may be confounded by factors such as previous immunosuppressant therapy, potential cross-contaminant use of immunosuppression COVID-19 medications (e.g., combination therapies with disease-modifying anti-rheumatic drugs), differential COVID-19 treatment in fear of cross-contaminant drug use, and increased prevalence of comorbidities associated with severe infection (e.g., diabetes, obesity, older age) within the prior autoimmune disease population [50]. Our study was also comprehensive in employing both SNOMED and ICD10 codes, mitigating some of the bias that may depend on whether an ICD10 code was available and reducing the barrier for information sharing between registries that used only SNOMED or only ICD10 codes. The robustness of our findings thus strengthens the plausibility of the association between increased COVID-19 severity and increased risk of new autoimmune disease.

While our study provides valuable insights, it is important to acknowledge its limitations. Given the lack of test supply early in the pandemic, testing for SARS-CoV-2 infection in the U.S. was initially reserved for hospitalized patients, while those with presumed SARS-CoV-2 infection not sick enough to require hospitalization were urged to quarantine at home contributing to type 2 error by not counting true cases of COVID-19. To mitigate the possibility of misclassification, we included a clinical diagnosis of COVID-19. Recent studies have also demonstrated that relying on polymerase chain reaction (PCR)-confirmed or PCR or antigen test-confirmed infections may not capture those in the community or those diagnosed by home testing or those who were felt to have classical chest x-ray findings of COVID-19 [4,15,17,22]. Although we employed two billing codes on separate days to identify incident autoimmune disease, rather than one as in most past studies, we acknowledge that no validated definition for all autoimmune disease exists or is feasible in such data, and it may have led to some misclassification of disease status. We began follow-up 30 days after the index SARS-CoV-2 infection to reduce the likelihood that infection-related non-specific symptoms of systemic inflammation were miscoded as new autoimmune disease, but this may still have occurred. Another potential limitation is that two years of follow-up would not capture long-term risk of autoimmune disease; autoantibodies may be present for up to ten years prior to development of RA, for example [51]. Additionally, patients with autoimmune diseases often experience prolonged "diagnostic delays", both due to lack of recognition of classical signs and symptoms, and due to the slow onset of disease. Thus, it is likely that some cases of autoimmune disease were undiagnosed in these study cohorts. The broad nature of symptoms attributed to post-acute syndrome of COVID-19 (PASC) may have also led to underdiagnosis of autoimmune disease in individuals after COVID-19. However, this misclassification would likely have biased our results toward the null, if at all. Disparate findings between N3C and PCORnet with respect to COVID severity and risk of autoimmune disease may be due in part to inherent network differences. During the initial phases of COVID-19 pandemic, ICUs were at capacity and many smaller hospitals sought higher level care for the most seriously ill patients. Patients who were more ill may have been referred to larger clinical sites represented in PCORnet for their COVID-19 care, but not followed there before or after, which may explain some of the discrepancies seen between the N3C and PCORnet. Patients in N3C had more encounters in their EHR and higher baseline healthcare utilization than did those enrolled in PCORnet; incomplete capture of baseline covariates and outcomes may have influenced results in PCORnet more than the other networks. Additionally, despite the discrepancy COVID-19 severity levels aHRs, we identified prior utilization had an association with both COVID-19 severity and incident AD with risk of incident AD increasing as prior utilization increases across networks (S5 Fig). Unfortunately,

prior utilization is especially sensitive to data warehouse constructions and architecture, and this may have contributed to disparate results seen across networks.

While we adjusted for all important confounders identified, there was much missing data on some important lifestyle factors, such as that smoking, which can both increase COVID-19 severity and risk factor for autoimmune conditions such as rheumatoid arthritis [52–54] Thus, there is the possibility of both misclassification and residual confounding by such shared risk factors. We were also unable to include vaccination status as immunization was not well documented in the RECOVER EHR cohort, and therefore strategically selected an enrollment period a priori so that vaccination use would naturally be limited in the study population. However, we could not fully assess the role of vaccination on AD incidence during 2 years of follow-up. A subsequent check of raw vaccination numbers confirmed that <1% of the population was vaccinated with any COVID-19 vaccine in the 180–14 days prior of their index SARS-CoV-2 infection across EHR networks. Due to our study period, we were unable to assess the complex relationship between vaccination, autoimmune disease, and COVID-19 that future research must address. Specifically, the potential for vaccination to both mitigate (through SARS-CoV-2 prevention) or increase autoimmune disease risk via mechanisms such as molecular mimicry, adjuvants, and bystander activation requires further study [55]. Our study design was also not intended to investigate the impact of reinfections after the index infection or other putative effect modifiers on autoimmune disease outcomes. Furthermore, we performed analyses adjusting for multiple potential confounders but were unable to perform propensity score matched or weighted analyses given relatively small sample sizes in each of the categories of infection, particularly among the PEDS-net population.

## Conclusion

The current study presents compelling evidence of an association of increasing acute COVID-19 severity with increasing risk of developing autoimmune diseases in the period up to two years following infection. Ongoing data collection and analyses of post-COVID-19 syndromes and autoimmune disease at the population level may be able to elucidate relationships of incident autoimmune disease with different COVID-19 variants, and the influence of having more than one SARS-CoV-2 infection. An equally important question for future studies is whether primary SARS-CoV-2 infection after April 1, 2021, potentially with less virulent viral strains that began to circulate in late 2021, along with increasing population vaccination, are associated with lower future risks of post-COVID-19 autoimmune disease.

## Supporting information

**S1 Appendix. PCORI RECOVER EHR selection criteria.**
(PDF)

**S2 Appendix. Autoimmune disease concept set groupings.**
(XLSX)

**S3 Appendix. ICD-9 CM, ICD-10 CM autoimmune disease concept codes.**
(XLSX)

**S4 Appendix. SNOMED autoimmune disease concept codes.**
(XLSX)

**S5 Appendix. ICD-9 CM, ICD-10 CM, and SNOMED substance use disorder concept codes.**
(XLSX)

**S6 Appendix. Systemic glucocorticoid RxNorm codes.**
(CSV)

**S7 Appendix.   COVID-19 severity indicator codes used in N3C.**
(XLSX)

**S8 Appendix.   COVID-19 severity indicatory codes used in PCORnet.**
(XLSX)

**S9 Appendix.   Incidence rates per 1000 –person-years among overall populations for all autoimmune disease concepts by EHR network.**
(XLSX)

**S1 Table.   PEDSnet severity concepts for mild- severe severity definitions.** Table from Rao et al. eClinicalMedicine. (2025) 80 *in Press* [1]; Forrest et al. Pediatrics (2022) 149 (4): e2021055765 [2].
(DOCX)

**S2 Table.   N3C multivariable models (1–7)- aHRs for incident autoimmune disease for all covariates by COVID-19 severity categories from day > 30 to end of study period.**
(DOCX)

**S3 Table.   PCORnet multivariable models (1–7)- aHRs for incident autoimmune disease for all covariates by COVID-19 severity categories from day > 30 to end of study period.**
(DOCX)

**S4 Table.   PEDSnet multivariable models (1–7)- aHRs for incident autoimmune disease for all covariates by COVID-19 severity categories from day > 30 to end of study period.**
(DOCX)

**S1 Fig.   N3C cumulative incidence curve for any new incident autoimmune disease.** N3C cumulative incidence curve for any incident autoimmune disease with competing risk of death by COVID-19 severity level with 95% CIs. Abbreviations: OP, outpatient; ED, emergency department; IP, inpatient(hospitalized); Vent, hospitalized and on ventilator.
(TIFF)

**S2 Fig.   PCORnet cumulative incidence curve for any new incident autoimmune disease.** PCORnet cumulative incidence curve for any incident autoimmune disease with competing risk of death by COVID-19 severity level with 95% CIs. Abbreviations: OP, outpatient; ED, emergency department; IP, inpatient(hospitalized); Vent, hospitalized and on ventilator.
(TIFF)

**S3 Fig.   PEDSnet cumulative incidence curve for any new incident autoimmune disease.** PEDSnet cumulative incidence curve for any incident autoimmune disease with competing risk of death by COVID-19 severity level with 95% CIs.
(TIF)

**S4 Fig.   Adjusted hazard ratios (95%CI) for any incident autoimmune disease by COVID-19 severity for N3C, PCORnet, and PEDSnet.** Forest plot of aHRs (95% CIs) for all regression models (1–7) for any incident autoimmune disease by COVID-19 severity levels across EHR networks.
(TIFF)

**S5 Fig.   Adjusted hazard ratios (95% CI) for any incident autoimmune disease by prior healthcare utilization level for N3C, PCORnet, and PEDsnet.** Forest plot of aHRs (95% CIs) for all regression models (1–7) for any incident autoimmune disease by prior healthcare utilization levels across EHR networks.
(TIFF)

## Acknowledgments

We would like to thank the National Community Engagement Group (NCEG), all patient, caregiver and community representatives, and all the participants enrolled in the RECOVER Initiative.

*We gratefully acknowledge the following core contributors to N3C:*

Adam B. Wilcox, Adam M. Lee, Alexis Graves, Alfred (Jerrod) Anzalone, Amin Manna, Amit Saha, Amy Olex, Andrea Zhou, Andrew E. Williams, Andrew Southerland, Andrew T. Girvin, Anita Walden, Anjali A. Sharathkumar, Benjamin Amor, Benjamin Bates, Brian Hendricks, Brijesh Patel, Caleb Alexander, Carolyn Bramante, Cavin Ward-Caviness, Charisse Madlock-Brown, Christine Suver, Christopher Chute, Christopher Dillon, Chunlei Wu, Clare Schmitt, Cliff Takemoto, Dan Housman, Davera Gabriel, David A. Eichmann, Diego Mazzotti, Don Brown, Eilis Boudreau, Elaine Hill, Emily Carlson Marti, Emily R. Pfaff, Evan French, Farrukh M Koraishy, Federico Mariona, Fred Prior, George Sokos, Greg Martin, Harold Lehmann, Heidi Spratt, Hemalkumar Mehta, J.W. Awori Hayanga, Jami Pincavitch, Jaylyn Clark, Jeremy Richard Harper, Jessica Islam, Jin Ge, Joel Gagnier, Johanna Loomba, John Buse, Jomol Mathew, Joni L. Rutter, Julie A. McMurry, Justin Guinney, Justin Starren, Karen Crowley, Katie Rebecca Bradwell, Kellie M. Walters, Ken Wilkins, Kenneth R. Gersing, Kenrick Dwain Cato, Kimberly Murray, Kristin Kostka, Lavance Northington, Lee Allan Pyles, Lesley Cottrell, Lili Portilla, Mariam Deacy, Mark M. Bissell, Marshall Clark, Mary Emmett, Matvey B. Palchuk, Melissa A. Haendel, Meredith Adams, Meredith Temple-O'Connor, Michael G. Kurilla, Michele Morris, Nasia Safdar, Nicole Garbarini, Noha Sharafeldin, Ofer Sadan, Patricia A. Francis, Penny Wung Burgoon, Philip R.O. Payne, Randeep Jawa, Rebecca Erwin-Cohen, Rena Patel, Richard A. Moffitt, Richard L. Zhu, Rishi Kamaleswaran, Robert Hurley, Robert T. Miller, Saiju Pyarajan, Sam G. Michael, Samuel Bozzette, Sandeep Mallipattu, Satyanarayana Vedula, Scott Chapman, Shawn T. O'Neil, Soko Setoguchi, Stephanie S. Hong, Steve Johnson, Tellen D. Bennett, Tiffany Callahan, Umit Topaloglu, Valery Gordon, Vignesh Subbian, Warren A. Kibbe, Wenndy Hernandez, Will Beasley, Will Cooper, William Hillegass, Xiaohan Tanner Zhang. Details of contributions available at covid.cd2h.org/core-contributors.

*The following N3C institutions whose data is released or pending:*

Adam B. Wilcox, Adam M. Lee, Alexis Graves, Alfred (Jerrod) Anzalone, Amin Manna, Amit Saha, Amy Olex, Andrea Zhou, Andrew E. Williams, Andrew Southerland, Andrew T. Girvin, Anita Walden, Anjali A. Sharathkumar, Benjamin Amor, Benjamin Bates, Brian Hendricks, Brijesh Patel, Caleb Alexander, Carolyn Bramante, Cavin Ward-Caviness, Charisse Madlock-Brown, Christine Suver, Christopher Chute, Christopher Dillon, Chunlei Wu, Clare Schmitt, Cliff Takemoto, Dan Housman, Davera Gabriel, David A. Eichmann, Diego Mazzotti, Don Brown, Eilis Boudreau, Elaine Hill, Emily Carlson Marti, Emily R. Pfaff, Evan French, Farrukh M Koraishy, Federico Mariona, Fred Prior, George Sokos, Greg Martin, Harold Lehmann, Heidi Spratt, Hemalkumar Mehta, J.W. Awori Hayanga, Jami Pincavitch, Jaylyn Clark, Jeremy Richard Harper, Jessica Islam, Jin Ge, Joel Gagnier, Johanna Loomba, John Buse, Jomol Mathew, Joni L. Rutter, Julie A. McMurry, Justin Guinney, Justin Starren, Karen Crowley, Katie Rebecca Bradwell, Kellie M. Walters, Ken Wilkins, Kenneth R. Gersing, Kenrick Dwain Cato, Kimberly Murray, Kristin Kostka, Lavance Northington, Lee Allan Pyles, Lesley Cottrell, Lili Portilla, Mariam Deacy, Mark M. Bissell, Marshall Clark, Mary Emmett, Matvey B. Palchuk, Melissa A. Haendel, Meredith Adams, Meredith Temple-O'Connor, Michael G. Kurilla, Michele Morris, Nasia Safdar, Nicole Garbarini, Noha Sharafeldin, Ofer Sadan, Patricia A. Francis, Penny Wung Burgoon, Philip R.O. Payne, Randeep Jawa, Rebecca Erwin-Cohen, Rena Patel, Richard A. Moffitt, Richard L. Zhu, Rishi Kamaleswaran, Robert Hurley, Robert T. Miller, Saiju Pyarajan, Sam G. Michael, Samuel Bozzette, Sandeep Mallipattu, Satyanarayana Vedula, Scott Chapman, Shawn T. O'Neil, Soko Setoguchi, Stephanie S. Hong, Steve Johnson, Tellen D. Bennett, Tiffany Callahan, Umit Topaloglu, Valery Gordon, Vignesh Subbian, Warren A. Kibbe, Wenndy Hernandez, Will Beasley, Will Cooper, William Hillegass, Xiaohan Tanner Zhang. Details of contributions available at covid.cd2h.org/core-contributors.

*Data Partners with Released Data:*

*The following institutions whose data is released or pending:*

Available: Advocate Health Care Network — UL1TR002389: The Institute for Translational Medicine (ITM) • Aurora Health Care Inc — UL1TR002373: Wisconsin Network For Health Research • Boston University Medical Campus — UL1TR001430: Boston University Clinical and Translational Science Institute • Brown University — U54GM115677: Advance Clinical Translational Research (Advance-CTR) • Carilion Clinic — UL1TR003015: iTHRIV Integrated Translational health Research Institute of Virginia • Case Western Reserve University — UL1TR002548: The Clinical & Translational Science Collaborative of Cleveland (CTSC) • Charleston Area Medical Center — U54GM104942: West Virginia Clinical and Translational Science Institute (WVCTSI) • Children's Hospital Colorado — UL1TR002535: Colorado Clinical and Translational Sciences Institute • Columbia University Irving Medical Center — UL1TR001873: Irving Institute for Clinical and Translational Research • Dartmouth College — None (Voluntary) Duke University — UL1TR002553: Duke Clinical and Translational Science Institute • George Washington Children's Research Institute — UL1TR001876: Clinical and Translational Science Institute at Children's National (CTSA-CN) • George Washington University — UL1TR001876: Clinical and Translational Science Institute at Children's National (CTSA-CN) • Harvard Medical School — UL1TR002541: Harvard Catalyst • Indiana University School of Medicine — UL1TR002529: Indiana Clinical and Translational Science Institute • Johns Hopkins University — UL1TR003098: Johns Hopkins Institute for Clinical and Translational Research • Louisiana Public Health Institute — None (Voluntary) • Loyola Medicine — Loyola University Medical Center • Loyola University Medical Center — UL1TR002389: The Institute for Translational Medicine (ITM) • Maine Medical Center — U54GM115516: Northern New England Clinical & Translational Research (NNE-CTR) Network • Mary Hitchcock Memorial Hospital & Dartmouth Hitchcock Clinic — None (Voluntary) • Massachusetts General Brigham — UL1TR002541: Harvard Catalyst • Mayo Clinic Rochester — UL1TR002377: Mayo Clinic Center for Clinical and Translational Science (CCaTS) • Medical University of South Carolina — UL1TR001450: South Carolina Clinical & Translational Research Institute (SCTR) • MITRE Corporation — None (Voluntary) • Montefiore Medical Center — UL1TR002556: Institute for Clinical and Translational Research at Einstein and Montefiore • Nemours — U54GM104941: Delaware CTR ACCEL Program • NorthShore University HealthSystem — UL1TR002389: The Institute for Translational Medicine (ITM) • Northwestern University at Chicago — UL1TR001422: Northwestern University Clinical and Translational Science Institute (NUCATS) • OCHIN — INV-018455: Bill and Melinda Gates Foundation grant to Sage Bionetworks • Oregon Health & Science University — UL1TR002369: Oregon Clinical and Translational Research Institute • Penn State Health Milton S. Hershey Medical Center — UL1TR002014: Penn State Clinical and Translational Science Institute • Rush University Medical Center — UL1TR002389: The Institute for Translational Medicine (ITM) • Rutgers, The State University of New Jersey — UL1TR003017: New Jersey Alliance for Clinical and Translational Science • Stony Brook University — U24TR002306 • The Alliance at the University of Puerto Rico, Medical Sciences Campus — U54GM133807: Hispanic Alliance for Clinical and Translational Research (The Alliance) • The Ohio State University — UL1TR002733: Center for Clinical and Translational Science • The State University of New York at Buffalo — UL1TR001412: Clinical and Translational Science Institute • The University of Chicago — UL1TR002389: The Institute for Translational Medicine (ITM) • The University of Iowa — UL1TR002537: Institute for Clinical and Translational Science • The University of Miami Leonard M. Miller School of Medicine — UL1TR002736: University of Miami Clinical and Translational Science Institute • The University of Michigan at Ann Arbor — UL1TR002240: Michigan Institute for Clinical and Health Research • The University of Texas Health Science Center at Houston — UL1TR003167: Center for Clinical and Translational Sciences (CCTS) • The University of Texas Medical Branch at Galveston — UL1TR001439: The Institute for Translational Sciences • The University of Utah — UL1TR002538: Uhealth Center for Clinical and Translational Science • Tufts Medical Center — UL1TR002544: Tufts Clinical and Translational Science Institute • Tulane University — UL1TR003096: Center for Clinical and Translational Science • The Queens Medical Center — None (Voluntary) • University Medical Center New Orleans — U54GM104940: Louisiana Clinical and Translational Science (LA CaTS) Center • University of Alabama at Birmingham — UL1TR003096: Center for Clinical and Translational Science • University of Arkansas for Medical Sciences — UL1TR003107: UAMS Translational Research Institute • University of Cincinnati — UL1TR001425: Center for Clinical and

Translational Science and Training • University of Colorado Denver, Anschutz Medical Campus — UL1TR002535: Colorado Clinical and Translational Sciences Institute • University of Illinois at Chicago — UL1TR002003: UIC Center for Clinical and Translational Science • University of Kansas Medical Center — UL1TR002366: Frontiers: University of Kansas Clinical and Translational Science Institute • University of Kentucky — UL1TR001998: UK Center for Clinical and Translational Science • University of Massachusetts Medical School Worcester — UL1TR001453: The UMass Center for Clinical and Translational Science (UMCCTS) • University Medical Center of Southern Nevada — None (voluntary) • University of Minnesota — UL1TR002494: Clinical and Translational Science Institute • University of Mississippi Medical Center — U54GM115428: Mississippi Center for Clinical and Translational Research (CCTR) • University of Nebraska Medical Center — U54GM115458: Great Plains IDeA-Clinical & Translational Research • University of North Carolina at Chapel Hill — UL1TR002489: North Carolina Translational and Clinical Science Institute • University of Oklahoma Health Sciences Center — U54GM104938: Oklahoma Clinical and Translational Science Institute (OCTSI) • University of Pittsburgh — UL1TR001857: The Clinical and Translational Science Institute (CTSI) • University of Pennsylvania — UL1TR001878: Institute for Translational Medicine and Therapeutics • University of Rochester — UL1TR002001: UR Clinical & Translational Science Institute • University of Southern California — UL1TR001855: The Southern California Clinical and Translational Science Institute (SC CTSI) • University of Vermont — U54GM115516: Northern New England Clinical & Translational Research (NNE-CTR) Network • University of Virginia — UL1TR003015: iTHRIV Integrated Translational health Research Institute of Virginia • University of Washington — UL1TR002319: Institute of Translational Health Sciences • University of Wisconsin-Madison — UL1TR002373: UW Institute for Clinical and Translational Research • Vanderbilt University Medical Center — UL1TR002243: Vanderbilt Institute for Clinical and Translational Research • Virginia Commonwealth University — UL1TR002649: C. Kenneth and Dianne Wright Center for Clinical and Translational Research • Wake Forest University Health Sciences — UL1TR001420: Wake Forest Clinical and Translational Science Institute • Washington University in St. Louis — UL1TR002345: Institute of Clinical and Translational Sciences • Weill Medical College of Cornell University — UL1TR002384: Weill Cornell Medicine Clinical and Translational Science Center • West Virginia University — U54GM104942: West Virginia Clinical and Translational Science Institute (WVCTSI) Submitted: Icahn School of Medicine at Mount Sinai —UL1TR001433: ConduITS Institute for Translational Sciences • The University of Texas Health Science Center at Tyler — UL1TR003167: Center for Clinical and Translational Sciences (CCTS) • University of California, Davis — UL1TR001860: UCDavis Health Clinical and Translational Science Center • University of California, Irvine — UL1TR001414: The UC Irvine Institute for Clinical and Translational Science (ICTS) • University of California, Los Angeles — UL1TR001881: UCLA Clinical Translational Science Institute • University of California, San Diego — UL1TR001442: Altman Clinical and Translational Research Institute • University of California, San Francisco — UL1TR001872: UCSF Clinical and Translational Science Institute NYU Langone Health Clinical Science Core, Data Resource Core, and PASC Biorepository Core — OTA-21-015A: Post-Acute Sequelae of SARS-CoV-2 Infection Initiative (RECOVER)Pending: Arkansas Children's Hospital — UL1TR003107: UAMS Translational Research Institute • Baylor College of Medicine — None (Voluntary) • Children's Hospital of Philadelphia — UL1TR001878: Institute for Translational Medicine and Therapeutics • Cincinnati Children's Hospital Medical Center — UL1TR001425: Center for Clinical and Translational Science and Training • Emory University — UL1TR002378: Georgia Clinical and Translational Science Alliance • HonorHealth — None (Voluntary) • Loyola University Chicago — UL1TR002389: The Institute for Translational Medicine (ITM) • Medical College of Wisconsin — UL1TR001436: Clinical and Translational Science Institute of Southeast Wisconsin • MedStar Health Research Institute — None (Voluntary) • Georgetown University — UL1TR001409: The Georgetown-Howard Universities Center for Clinical and Translational Science (GHUCCTS) • MetroHealth — None (Voluntary) • Montana State University — U54GM115371: American Indian/Alaska Native CTR • NYU Langone Medical Center — UL1TR001445: Langone Health's Clinical and Translational Science Institute • Ochsner Medical Center — U54GM104940: Louisiana Clinical and Translational Science (LA CaTS) Center • Regenstrief Institute — UL1TR002529: Indiana Clinical and Translational Science Institute • Sanford Research — None (Voluntary) • Stanford

University — UL1TR003142: Spectrum: The Stanford Center for Clinical and Translational Research and Education • The Rockefeller University — UL1TR001866: Center for Clinical and Translational Science • The Scripps Research Institute — UL1TR002550: Scripps Research Translational Institute • University of Florida — UL1TR001427: UF Clinical and Translational Science Institute • University of New Mexico Health Sciences Center — UL1TR001449: University of New Mexico Clinical and Translational Science Center • University of Texas Health Science Center at San Antonio — UL1TR002645: Institute for Integration of Medicine and Science • Yale New Haven Hospital — UL1TR001863: Yale Center for Clinical Investigation.

*We gratefully acknowledge sites and principal investigators within the PCORI Consortium:*

**Louisiana Public Health Institute:** *Tom Carton, mPI,* Anna Legrand, Elizabeth Nauman.

**Weill Cornell Medicine:** *Rainu Kaushal, mPI, Mark Weiner, mPI,* Sajjad Abedian, Dominique Brown, Christopher Cameron, Thomas Campion, Andrea Cohen, Marietou Dione, Rosie Ferris, Wilson Jacobs, Michael Koropsak, Alex LaMar, Colby V. Lewis, Dmitry Morozyuk, Peter Morrisey, Duncan Orlander, Jyotishman Pathak, Mahfuza Sabiha, Edward J. Schenck, Stephenson Strobel, Zoe Verzani, Fei Wang, Zhenxing Xu, Chengxi Zang, Yongkang Zhang.

**Children's Hospital of Philadelphia:** *L. Charles Bailey, mPI, Christopher B. Forrest, mPI,* Rodrigo Azuero-Dajud, Andrew Samuel Boss, Morgan Botdorf, Colleen Byrne, Peter Camacho, Abigail Case, Kimberley Dickinson, Susan Hague, Jonathan Harvell, Miranda Higginbotham, Kathryn Hirabayshi, Sandra Ilunga, Rochelle Jordan, Aqsa Khan, Vitaly Lorman, Nicole Marchesani, Sahal Master, Jill McDonald, Nhat Nguyen, Hanieh Razzaghi, Qiwei Shen, Alexander Shorrock, Levon H. Utidjian, Kaleigh Wieand.

**Children's Hospital of Colorado:** *Suchitra Rao, mPI.*

**PCORnet Data Contributors**

**Albert Einstein College of Medicine** *Parsa Mirhaji, PI* | **Ann & Robert H. Lurie Children's Hospital of Chicago** *Ravi Jhaveri, PI,* | **Children's Hospital of Philadelphia** *L. Charles Bailey, mPI, Christopher B. Forrest, mPI* | **Children's National Hospital** *Hiroki Morizono, PI* | **Cincinnati Children's Hospital Medical Center** *Nathan M. Pajor, PI* | **Columbia University** *Soumitra Sengupta, PI* | **Duke University Health System** *W. Schuyler Jones, PI,* Curtis Kieler | **Emory University** | **Feinberg School of Medicine, Northwestern University** *David M. Liebovitz, PI,* | **Icahn School of Medicine at Mount Sinai** *Carol R. Horowitz, PI,* Patricia Kovatch | **Intermountain Healthcare** Heidi T. May | **Intermountain Medical Center Heart Institute** *Benjamin D. Horne, PI* | **Medical College of Wisconsin** *Bradley Taylor, PI,* Alex Stoddard | **Medical University of South Carolina** | **Nationwide Children's Hospital** *Kelly Kelleher, PI,* Yungui Huang | **Nemours/Alfred I. duPont Hospital for Children** *H. Timothy Bunnell, PI* | **Nicklaus Children's Hospital** *Sandy L. Gonzalez, PI,* Maurice Duque | **New York University Langone Health** *Saul Blecker, PI,* Nathalia Ladino | **OCHIN, Inc**. *Marion Ruth Sills, PI* | **Ochsner Health System** *Dan Fort, PI* | **Penn State University College of Medicine** *Cynthia H. Chuang, PI,* Wenke Hwang | **Seattle Children's Hospital** *Dimitri A. Christakis, PI,* Daksha Ranade | **Stanford University School of Medicine** *Hayden T. Schwenk, PI,* Keith E. Morse | **Temple University** *Sharon J. Herring, PI,* Aaron D. Mishkin | **The Ohio State University** Neena A. Thomas | **The Research Institute at Nationwide Children's Hospital** Yungui Huang | **University Medical Center New Orleans** *Yuriy Bisyuk, PI* | **University of California San Francisco** *Susan Kim, PI,* Mark J. Pletcher | **University of Colorado School of Medicine and Children's Hospital Colorado** *Suchitra Rao, mPI,* Sara J. Deakyne Davies | **University of Florida** *Mei Liu, PI* | **University of Iowa** *Elizabeth A. Chrischilles, PI,* Boyd M. Knosp | **University of Miami** | **University of Michigan** *David A. Williams, PI,* James B. Henderson | **University of Missouri School of Medic**ine *Abu Saleh Mohammad Mosa, PI,* Xing Song | **University of Nebraska Medical Center** *Carol Reynolds Geary, PI,* Jerrod Anzalone | **University of Pittsburgh** *Jonathan Arnold, PI, Michael J. Becich, PI* | **University of South Florida** | **University of Texas Southwestern Medical Center** *Lindsay G. Cowell, PI,* | **University of Utah** *Mollie R. Cummins, PI,* Ramkiran Gouripeddi | **Vanderbilt University Medical Center** *Yacob Tedla, PI,* | **Wake Forest School of Medicine** *Stephen M. Downs, PI,* Brian Ostasiewski | **Weill Cornell Medicine** *Rainu Kaushal, mPI,* Thomas Campion.

**Disclaimer**: Authorship was determined using ICMJE recommendations. The content is solely the responsibility of the authors and does not necessarily represent the official views of the National Institutes of Health, N3C, PCORI, or RECOVER. **N3C Disclaimer:** The N3C Publication Committee confirmed that this manuscript is in accordance with N3C data use and attribution policies; however, this content is solely the responsibility of the authors and does not necessarily represent the official views of the National Institutes of Health or N3C program.

## Author contributions

**Conceptualization:** Shannon Wuller, Nora G. Singer, Colby Lewis, Elizabeth W. Karlson, Grant S. Schulert, Jason D. Goldman, Jennifer Hadlock, Jonathan Arnold, Kathryn Hirabayashi, Lauren E. Stiles, Lawrence C. Kleinman, Lindsay G. Cowell, Mady Hornig, Margaret A. Hall, Mark G. Weiner, Michael Koropsak, Michelle F Lamendola-Essel, Rachel Kenney, Richard A. Moffitt, Sajjad Abedian, Shari Esquenazi-Karonika, Steven G. Johnson, Stephenson Stroebel, Zachary S. Wallace, Karen H. Costenbader.

**Data curation:** Kathryn Hirabayashi, Margaret A. Hall, Michael Koropsak, Sajjad Abedian, Stephenson Stroebel.

**Formal analysis:** Colby Lewis, Kathryn Hirabayashi, Margaret A. Hall.

**Methodology:** Shannon Wuller, Nora G. Singer, Colby Lewis, Elizabeth W. Karlson, Jason D. Goldman, Jennifer Hadlock, Jonathan Arnold, Kathryn Hirabayashi, Lawrence C. Kleinman, Mady Hornig, Mark G. Weiner, Michael Koropsak, Rachel Kenney, Zachary S. Wallace, Karen H. Costenbader.

**Project administration:** Shannon Wuller.

**Supervision:** Karen H. Costenbader.

**Visualization:** Shannon Wuller, Colby Lewis, Kathryn Hirabayashi, Margaret A. Hall.

**Writing – original draft:** Shannon Wuller, Nora G. Singer, Karen H. Costenbader.

**Writing – review & editing:** Shannon Wuller, Nora G. Singer, Colby Lewis, Elizabeth W. Karlson, Grant S. Schulert, Jason D. Goldman, Jennifer Hadlock, Jonathan Arnold, Kathryn Hirabayashi, Lauren E. Stiles, Lawrence C. Kleinman, Lindsay G. Cowell, Mady Hornig, Margaret A. Hall, Mark G. Weiner, Michael Koropsak, Michelle F Lamendola-Essel, Rachel Kenney, Richard A. Moffitt, Sajjad Abedian, Shari Esquenazi-Karonika, Steven G. Johnson, Stephenson Stroebel, Zachary S. Wallace, Karen H. Costenbader.

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
