## [Decision Letter · Decision Letter 0]

3 Jan 2025

PONE-D-24-35787Severity of acute SARS-CoV-2 infection and risk of new-onset autoimmune disease: A RECOVER initiative study in nationwide U.S. cohortsPLOS ONE

Dear Dr. Wuller,

Thank you for submitting your manuscript to PLOS ONE. After careful consideration, we feel that it has merit but does not fully meet PLOS ONE’s publication criteria as it currently stands. Therefore, we invite you to submit a revised version of the manuscript that addresses the points raised during the review process.

We look forward to receiving your revised manuscript.

Kind regards,

Dong Keon Yon, MD, FACAAI, FAAAAI

Academic Editor

PLOS ONE

**Journal Requirements:**

I have read the journal's policy and the authors of this manuscript have the following competing interests: JH has received funding paid to the Institute of Systems Biology from Bristol Myers Squibb, Gilead, Janssen, Novartis and Pfizer for research unrelated to this study.

We note that one or more of the authors are employed by a commercial company. 

“The funder provided support in the form of salaries for authors, but did not have any additional role in the study design, data collection and analysis, decision to publish, or preparation of the manuscript. The specific roles of these authors are articulated in the ‘author contributions’ section.”

3. Please upload a new copy of Figures 2A, 2B, S1, S2, S3, S4, S5 and S6 as the detail is not clear. Please follow the link for more information: "https://blogs.plos.org/plos/2019/06/looking-good-tips-for-creating-your-plos-figures-graphics/
https://blogs.plos.org/plos/2019/06/looking-good-tips-for-creating-your-plos-figures-graphics/"

**Additional Editor Comments:**

Thank you for submitting your manuscript. The reviewers and I believe it is of potential value for our readers. However, the reviewers have raised a number of very important issues, and their excellent comments will need to be adequately addressed in a revision before the acceptability of your manuscript for publication in the Journal can be determined. We cannot guarantee that your revised paper will be chosen for publication; this would be solely based on how satisfactorily you have addressed the reviewer comments.

Reviewers' comments:

Reviewer's Responses to Questions

**Comments to the Author**

1. Is the manuscript technically sound, and do the data support the conclusions?

Reviewer #1: Yes

Reviewer #2: Yes

2. Has the statistical analysis been performed appropriately and rigorously? 

Reviewer #1: Yes

Reviewer #2: Yes

3. Have the authors made all data underlying the findings in their manuscript fully available?

Reviewer #1: Yes

Reviewer #2: Yes

4. Is the manuscript presented in an intelligible fashion and written in standard English?

Reviewer #1: Yes

Reviewer #2: Yes

5. Review Comments to the Author

**Reviewer #1: ** The study "Severity of Acute SARS-CoV-2 Infection and Risk of New-Onset Autoimmune Disease" investigates how COVID-19 severity influences the risk of developing autoimmune diseases. The results indicated that severe COVID-19 cases had a significantly higher risk of new-onset autoimmune disease. In adults, diseases like inflammatory arthritis and Sjögren’s syndrome were common, while in children, conditions such as type 1 diabetes and hematological autoimmune diseases were observed. The highest severity category, "hospitalized with ventilation," was associated with a particularly elevated risk, underscoring a dose-response relationship between COVID-19 severity and autoimmune disease development.

Overall, this article well designed and written. I only have some MINOR comments:

1. Introduction Background. The introduction provides a solid foundation by discussing the potential link between viral infections and autoimmunity. To enhance this section, a brief overview of existing literature on COVID-19's role in triggering autoimmune responses would contextualize the study's significance. This addition would underscore the relevance of investigating the association between COVID-19 severity and subsequent autoimmune disease onset.

2. Justification of Disease Classification. The study defines new-onset autoimmune disease based on the presence of two new ICD codes within 30 days. While this criterion is practical, it may not fully account for transient inflammatory responses that do not culminate in chronic autoimmune conditions. Providing a rationale for this definition or acknowledging it as a limitation would strengthen the study's methodological transparency.

3. Adjustment for Covariates. The multivariable models adjust for several baseline characteristics, which is commendable. However, the discussion could be enriched by exploring whether factors such as body mass index (BMI), socioeconomic status, or ethnicity might confound the observed association between COVID-19 severity and autoimmune risk. Addressing these variables would provide a more comprehensive understanding of potential confounders.

4. Inclusion of Vaccination Status. Considering the role of COVID-19 vaccination in potentially mitigating autoimmune risk, it would be pertinent to mention whether participants' vaccination statuses were considered or excluded. This information would enhance the study's relevance to current public health considerations.

5. Discussion. Suggest adding more discussion on literatures and mechanism and cite relevant references. eg. 6. https://onlinelibrary.wiley.com/doi/pdfdirect/10.1111/1756-185X.14963;
https://onlinelibrary.wiley.com/doi/epdf/10.1111/1756-185X.14766

**Reviewer #2: ** This study presents interesting findings, but there are a few concerns that need to be addressed.

1. The term COVID-19 infection is inaccurate. It is recommended to revise it to SARS-CoV-2 infection for precision.

2. Have you considered addressing reinfection in your analysis? Please clarify.

3. Regarding the phrase World Health Organization COVID-19 severity category for adults, while it is clear that you are using WHO guidelines, additional details are needed. Specifically, what codes were employed, and how was the time frame defined?

4. For the PEDSnet acute COVID-19 illness severity classification system for children, please provide more detailed information regarding its criteria and application.

5. Include the follow-up duration for each cohort used in the study.

6. The resolution of the figures is too low, rendering them uninterpretable and not informative. Kindly provide high-resolution versions of all figures.

7. It is suggested to refer to PMID: 38437702, not only for discussion but also as a reference for expressions such as SARS-CoV-2 infection.

6. PLOS authors have the option to publish the peer review history of their article (what does this mean? ). If published, this will include your full peer review and any attached files.

**Do you want your identity to be public for this peer review?** For information about this choice, including consent withdrawal, please see our Privacy Policy .

Reviewer #1: No

Reviewer #2: No

---

## [Author Response · Author response to Decision Letter 1]

22 Apr 2025

Thank you for the opportunity for to revise and resubmit. Our updated manuscript has been changed to align with reviewer and editor comments, as stated in the decision letter, and includes the following changes:

1. Updated formatting to align with PLOS ONE

2. Updating competing interest and funding statement within the Cover letter

3. Uploaded new figures and supplementals for clarity

4. Provided a Response to reviewer rebuttal letter addressing reviewer comments

5. provided a tracked changes manuscript

6. provided a cleaned version of the updated manuscript

---

## [Decision Letter · Decision Letter 1]

28 Apr 2025

Severity of acute SARS-CoV-2 infection and risk of new-onset autoimmune disease: A RECOVER initiative study in nationwide U.S. cohorts

PONE-D-24-35787R1

Dear Dr. Wuller,

We’re pleased to inform you that your manuscript has been judged scientifically suitable for publication and will be formally accepted for publication once it meets all outstanding technical requirements.

Kind regards,

Dong Keon Yon, MD, FACAAI, FAAAAI

Academic Editor

PLOS ONE

Additional Editor Comments (optional):

This is an excellent paper!

Reviewers' comments:

Reviewer's Responses to Questions

**Comments to the Author**

1. If the authors have adequately addressed your comments raised in a previous round of review and you feel that this manuscript is now acceptable for publication, you may indicate that here to bypass the “Comments to the Author” section, enter your conflict of interest statement in the “Confidential to Editor” section, and submit your "Accept" recommendation.

Reviewer #1: All comments have been addressed

Reviewer #2: All comments have been addressed

2. Is the manuscript technically sound, and do the data support the conclusions?

Reviewer #1: Yes

Reviewer #2: Yes

3. Has the statistical analysis been performed appropriately and rigorously? 

Reviewer #1: Yes

Reviewer #2: Yes

4. Have the authors made all data underlying the findings in their manuscript fully available?

Reviewer #1: Yes

Reviewer #2: Yes

5. Is the manuscript presented in an intelligible fashion and written in standard English?

Reviewer #1: Yes

Reviewer #2: Yes

6. Review Comments to the Author

Reviewer #1: All previous comments were answered.

I have no further comment. No additional comments for the author, including concerns about dual publication, research ethics, or publication ethics.

Reviewer #2: (No Response)

7. PLOS authors have the option to publish the peer review history of their article (what does this mean? ). If published, this will include your full peer review and any attached files.

**Do you want your identity to be public for this peer review?** For information about this choice, including consent withdrawal, please see our Privacy Policy .

Reviewer #1: No

Reviewer #2: No

---

## [Editor Report · Acceptance letter]

PONE-D-24-35787R1

PLOS ONE

Dear Dr. Wuller,

I'm pleased to inform you that your manuscript has been deemed suitable for publication in PLOS ONE. Congratulations! Your manuscript is now being handed over to our production team.

Kind regards,

on behalf of

Dr. Dong Keon Yon

Academic Editor

PLOS ONE